# Cellular geometry scaling ensures robust division site positioning

Ying Gu[1,2] & Snezhana Oliferenko [1,2]

Cells of a specific cell type may divide within a certain size range. Yet, functionally optimal cellular organization is typically maintained across different cell sizes, a phenomenon known as scaling. The mechanisms underlying scaling and its physiological significance remain elusive. Here we approach this problem by interfering with scaling in the rod-shaped fission yeast *Schizosaccharomyces japonicus* that relies on cellular geometry cues to position the division site. We show that *S. japonicus* uses the Cdc42 polarity module to adjust its geometry to changes in the cell size. When scaling is prevented resulting in abnormal cellular length-to-width aspect ratio, cells exhibit severe division site placement defects. We further show that despite the generally accepted view, a similar scaling phenomenon can occur in the sister species, *Schizosaccharomyces pombe*. Our results demonstrate that scaling is required for normal cell function and delineate possible rules for cellular geometry maintenance in populations of proliferating cells.

[1] The Francis Crick Institute, 1 Midland Road, London NW1 1AT, UK. [2] Randall Centre for Cell and Molecular Biophysics, School of Basic and Medical Biosciences, King's College London, London SE1 1UL, UK. Correspondence and requests for materials should be addressed to S.O. (email: snezhka.oliferenko@crick.ac.uk)

Cell size homeostasis in proliferating cells is achieved by balancing the rates of growth and division[1–9]. The extent of growth is influenced by the metabolic activity that fuels cellular biosynthetic machineries[10–12]. As nutrient supply and other environmental factors bear on cellular metabolism and ultimately growth rate, proliferating populations may settle on different cell size optima depending on the circumstances[13–17]. Both cell size and shape affect cellular surface to volume ratio impacting on nutrient exchange and ion and water fluxes[18,19]. Cellular geometry also influences the organization of cytoskeletal networks, the efficiency of diffusive and directional transport and establishment and function of intracellular gradients[18,19]. As proliferating polarized cells set a new threshold for cell size at division, cell shape may need to be scaled accordingly to sustain cell function[18,20,21].

The fission yeast Schizosaccharomyces japonicus (S. japonicus) is an emerging genetically tractable model system[22,23]. Its sister species Schizosaccharomyces pombe (S. pombe) has been long used to understand the relationship between cell growth, cell polarity and cell division[24–26]. Similar to the rod-shaped S. pombe, S. japonicus is a walled single-celled organism that grows at the cell tips and divides in the middle. Yet, S. japonicus biology is sufficiently different. Unlike S. pombe, it breaks down the nuclear envelope during mitosis[27,28] and does not use the dominant anillin Mid1-dependent pathway to assemble the actomyosin cytokinetic ring in early mitosis[23,29]. In fact, Mid1 dependency in S. pombe likely resulted from the fission yeast clade-specific anillin gene duplication followed by subfunctionalization of Mid1 orthologs. S. japonicus assembles the medial ring only after the exit from mitosis, similar to animal cells. To do so, it uses the Cdc15-dependent ring anchorage system relying on cell tip-localized cortical cues including the kinase Pom1, which appears to be ancestral within the fission yeast clade[23,29].

When advanced into mitosis due to premature Cdk1 activation, S. pombe cells divide at a shorter length, exhibiting a so-called wee phenotype[30–33]. This reduced cellular length-to-width aspect ratio in temperature-sensitive mutants in Cdk1 activation pathway indicates that, at least under these circumstances, S. pombe does not scale its geometry to cell volume. Cellular fitness is decreased as stumpy wee1 mutant cells exhibit inaccurate division site positioning following the upshift to the restrictive temperature, although the severity of the defects is buffered by the presence of the two actomyosin ring positioning pathways[34]. Intuitively, a system reliant solely on inhibiting ring assembly at the cell tips may not be robust to changes in cellular aspect ratio. As cells become shorter while maintaining the same width, and hence, the size of the polar zones, the cortical gradients of factors preventing ring assembly at the cell tips may become progressively shallower, encroaching into the equatorial cortex. We set out to test the robustness of this cell division strategy by attempting to manipulate the length-to-width aspect ratio in S. japonicus cells.

Our results show that the cellular aspect ratio indeed controls the fidelity of division site positioning in S. japonicus. Surprisingly, this organism can adjust its width to cell volume at division, preventing catastrophic loss of cellular geometry upon changes in nutritional or cell cycle cues. We further show that S. pombe is, in fact, capable of geometry scaling, although to a lesser extent.

## Results

**S. japonicus scales its geometry to changes in cell volume.** Advancing S. pombe cells into mitosis by inhibition of the tyrosine kinase Wee1 is thought to provide a straightforward way to decrease cellular length-to-width aspect ratio[30]. We decided to use this genetic approach to generate shorter S. japonicus cells. To this end, we engineered an ATP analog-sensitive allele of S. japonicus, wee1-as8, based on an established S. pombe version[35]. After treatment of asynchronous wee1-as8 populations with 20 μM ATP analog 3-BrB-PP1, cells first divided medially, albeit at a shorter length. As these short cells entered the next mitosis, their daughters assumed asymmetric pattern of growth. Whereas most of the cell cortex underwent transient isotropic growth, one of the cell tips hyperpolarized and grew out at a smaller diameter (Fig. 1a, b; see time-lapse images in Supplementary Fig. 1a). The next division typically occurred close to the neck of the pear-shaped cell. Following cytokinesis, the asymmetrically dividing cell produced a thinner daughter with scaled geometry that resumed symmetric divisions and a wider one that usually underwent another round of hyperpolarization and asymmetric division. The accuracy of division site positioning in terms of pole-to-pole distance in asymmetrically dividing cells remained comparable to control (Supplementary Fig. 1b). After a few cell cycles, the population of exponentially dividing wee1-as8 S. japonicus cells reset cellular length-to-width aspect ratio, with cells dividing at both smaller length and width (Fig. 1a, c). Upon reaching steady state, 3-BrB-PP1-treated wee1-as8 cells divided at 72% volume as compared to the solvent control ($284.4 \ \mu m^3 \pm 42.5 \ \mu m^3$ in 3-BrB-PP1-treated wee1-as8, $393.8 \ \mu m^3 \pm 30.73 \ \mu m^3$ in control, $n = 18$).

The reduction of cell width in Wee1-inhibited cells was fully reversible. When 3-BrB-PP1 was washed out, small wee1-as8 cells reverted to wild type size, by extending cell length at division and concomitantly increasing cell diameter, likely through partial depolarization of growth machinery (Fig. 1d, e).

We noted that 3-BrB-PP1 treatment also promoted hyperpolarization of growth in wild type S. japonicus, although to a lesser extent as compared to wee1-as8 cells ($p < 0.0001$ for all measured parameters, Kolmogorov–Smirnov tests), indicating off-target effects (Fig. 1b, c and Supplementary Fig. 1c, d, see also ref. [36]). This did not affect division plane positioning (Supplementary Fig. 1e). To confirm that specific downregulation of Wee1 kinase activity contributed to the morphological changes in 3-BrB-PP1-treated wee1-as8 cells, we constructed an S. japonicus version of wee1-50 S. pombe allele (G788E amino acid substitution), where Wee1 is functional at 24 °C but loses activity upon temperature up-shift[30]. Of note, S. japonicus cells harboring the wee1-G788E mutation divided at both reduced length and width when grown already at 30 °C. The extent of cell width reduction was lower as compared to 3-BrB-PP1 treatment of wee1-as8 cells, resulting in a slight deviation from the wild-type aspect ratio (Fig. 1f, g, see wild type images in Supplementary Fig. 1f). S. japonicus cells advanced into mitosis due to conditional G146D mutation in Cdc2 (cdc2-1w) that renders CDK1 insensitive to Wee1 inhibition[31–33], exhibited reduced cellular length and width already at 24 °C, with cell width decreasing further after a prolonged incubation at 36 °C (Supplementary Fig. 1h, i). In these cells, cellular length-to-width ratio was not maintained perfectly but decreased to approximately 2.5 (Supplementary Fig. 1i). Upon shift to the restrictive temperature, cells of both genotypes settled on new geometry parameters through an asymmetrically dividing stage (Supplementary Fig. 1g, h; 3-h time points).

Suggesting that the cellular aspect ratio scaling is a physiological response, S. japonicus prototrophic wild type cells underwent similar morphological transition upon shift from the nutrition-rich medium (3% glucose YE) to the chemically defined minimal medium (EMM) (Fig. 1h, i). Similarly, shifting cells to low glucose medium (0.2% glucose YE) also lead to decrease in cell length and width. The lower aspect ratio in cells grown in 0.2% glucose was likely due to glucose exhaustion in batch cultures, resulting in high phenotypic variability (Supplementary Fig. 1j, k). Overall, our results suggested that forcing S. japonicus

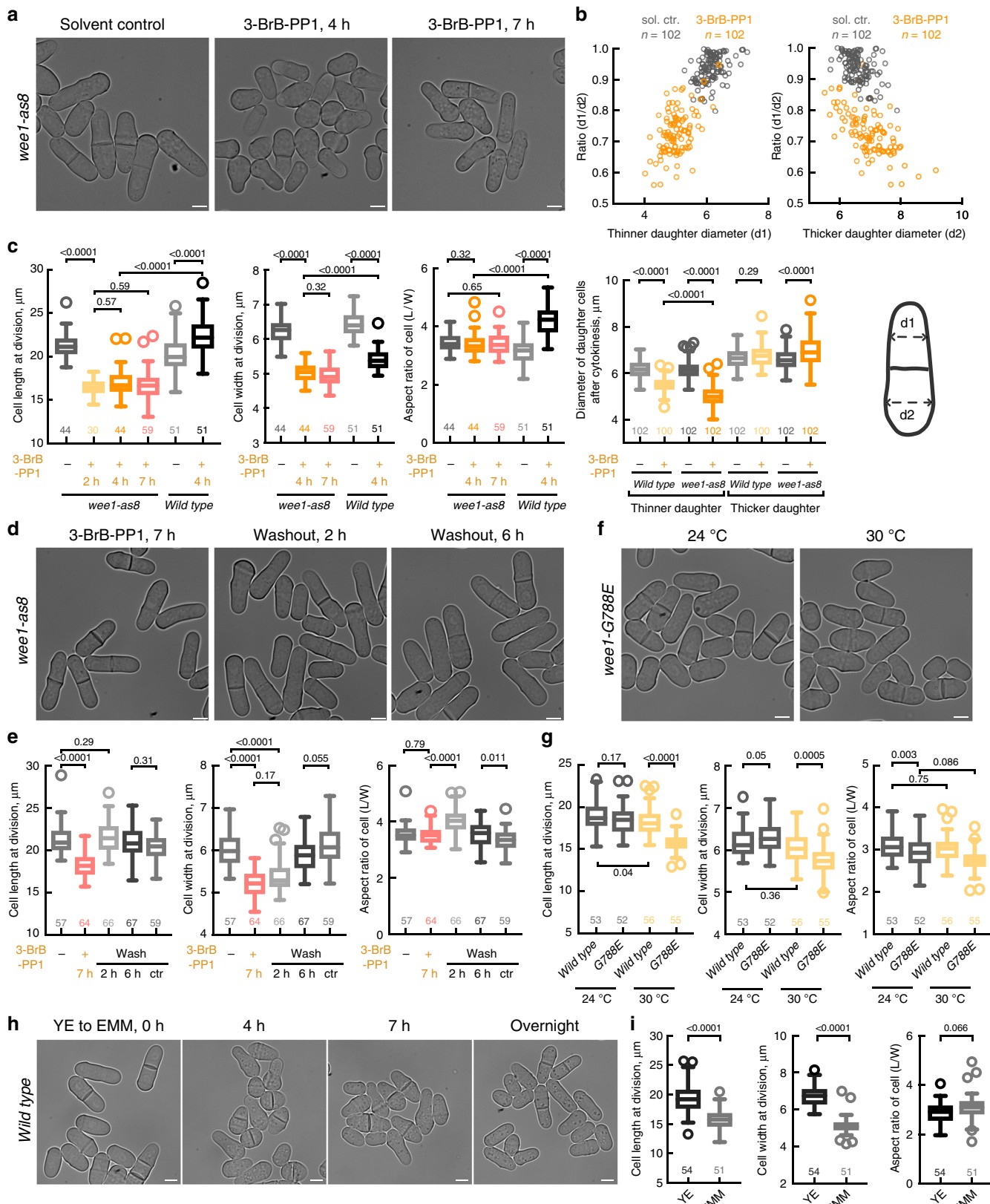

cells to divide at smaller volume either by manipulating Cdk1 activation status or nutritional availability leads to re-scaling of cellular geometry. Presumably, maintenance of the length-to-width aspect ratio may allow *S. japonicus* to adjust the spatial patterning of division site positioning determinants across a range of cellular volumes.

**A Cdc42 GAP Rga4 is required for cellular geometry scaling.** Reasoning that such a transition in growth patterns may require dynamic regulation of the small GTPase Cdc42 that is known to control polarized growth in both budding and fission yeasts[37,38], we used CRIB-3xGFP as an established fluorescent marker for Cdc42 activation[39]. Time-lapse sequences of CRIB-3xGFP-

**Fig. 1** *S. japonicus* maintains cellular aspect ratio over a range of volumes. **a** *S. japonicus* analog-sensitive *wee1-as8* cells incubated with methanol (solvent control) or 20 μM ATP analog 3-BrB-PP1. Note the morphological transition in 3-BrB-PP1-treated cells occurring at a 4-h time point. **b** Wee1 inhibition initially causes differences in the diameters of two daughter cells (orange circles indicate cells treated with 3-BrB-PP1 for 2 h; gray circles represent solvent control). Shown are scatter plots, where either "thinner" (top left) or "thicker" (top right) daughter cell diameter measurements are on *x*-axis, and the ratios between diameters of the daughters are on *y*-axis. Graph (bottom right) compares cell diameter changes of the two daughters between control and 3-BrB-PP1-treated *wee1-as8* and wild type populations. **c** Quantifications of cell length, width and aspect ratio at division of *wee1-as8* cells shown in (**a**) and similarly treated wild type cells shown in Supplementary Fig. 1c. **d** Wee1-inhibited cells recover their original dimensions following the removal of the ATP analog from the growth medium. Shown are *wee1-as8* cells treated with 20 μM 3-BrB-PP1 for 7 h, following the washout of the drug for 2 and 6 h. **e** Quantifications of cell length, width and aspect ratio at division of cells shown in (**d**). Methanol-treated *wee1-as8* cells washed with growth medium (5th column) were used to control for cell number increase and washing procedure. **f** *wee1-G788E* mutant cells cultured at 24 °C (left) and 30 °C (right) overnight. **g** Measurements of cellular length, width and aspect ratio of the wild type as compared to *wee1-G788E* cells shown in (**f**). **h** *S. japonicus* cells after the shift from YE to EMM for 0, 4, 7 h and overnight. **i** Quantifications of cell length, width and aspect ratio at division before and after shift from YE to EMM overnight, for cells shown in (**h**). **a**, **d**, **f**, **h** Shown are single *z*-plane bright-field micrographs of live cells; scale bars represent 5 μm.
**b**, **c**, **e**, **g**, **i** Quantifications presented as box plots with whiskers calculated by the Tukey method; *n* indicated in figures; *p* values derived from Kolmogorov–Smirnov test

expressing interphase *S. japonicus* cells revealed relatively weak and highly dynamic localization of active Cdc42 to the cell tips (Fig. 2a, b). This was in contrast to an extremely polarized distribution of active Cdc42 in *S. pombe* (Fig. 2a, b and ref. [40]). Estimations of the full width of the CRIB-3xGFP domain at half maximum (FWHM) in both species using Gaussian curve fit[41] and normalizing this values to cell tip radii showed that this marker of Cdc42 activity explores a broader area around the cell tips in *S. japonicus* as compared to its sister species (Fig. 2b, bottom panel). CRIB-3xGFP did not show higher enrichment at cell tips in *S. japonicus* cells that stabilized their geometry following long-term Wee1 inhibition or growth in the minimal medium (Supplementary Fig. 2a).

Of note, when *S. japonicus* cells initiated hyperpolarized growth soon after Wee1 inactivation, CRIB-3xGFP became more enriched at the growing cell tips (Fig. 2c, d; see relative tip intensity values in Supplementary Fig. 2). At this time point, the Cdc42 scaffold protein Scd2-mNeonGreen was also enriched at the hyperpolarized cell tips in Wee1-inhibited cells (Fig. 2c, d and Supplementary Fig. 2). Although 3-BrB-PP1 treatment also caused hyperpolarization in wild type cells, we did not detect enrichment of these markers of Cdc42 activity under these conditions (Fig. 2c, d and Supplementary Fig. 2d). Consistently, the density of actin cytoskeletal assemblies required for polarized cell growth, including actin cables and endocytic actin patches, also increased upon Wee1 inhibition (Supplementary Fig. 2b), and the cell polarity protein Tea4 became enriched at the hyperpolarized cell tips (Supplementary Fig. 2c).

The mNeonGreen-tagged Cdc42 guanine nucleotide exchange factor (GEF) Gef1 exhibited weak accumulation at the cell tips in control cells. The intensity of Gef1 increased slightly upon 3-BrB-PP1 treatment in wild type cells, rising further at the growing cell tips during morphological transition in Wee1-inhibited cells (Fig. 2c, d and Supplementary Fig. 2d). We did not detect any cortical enrichment of another Cdc42 GEF, Scd1-mNeonGreen, in either of these conditions (Fig. 2c, d and Supplementary Fig. 2d). Curiously, the Cdc42 GTPase activating protein (GAP) Rga4 covered virtually an entire cortex of Wee1-inhibited small cells with an exception of the narrow growing tip (Fig. 2c, d). The second Cdc42 GAP Rga6 (ref. [42] and Supplementary Fig. 2e) was spread broadly throughout the cortex in both control and Wee1-inhibited *S. japonicus* cells (Fig. 2c, d).

We concluded that whereas both fission yeast species maintain comparable rod-shaped morphology, the Cdc42-dependent growth machinery at steady state is considerably less polarized in *S. japonicus* as compared to *S. pombe*. Yet, the former species can potentially modify its polarity apparatus when scaling cellular aspect ratio with changes in the cell volume.

We wondered whether interfering with Cdc42 regulation could prevent cellular geometry scaling in a population and, if so, what would be functional consequences of such a failure. To this end, we analyzed the physiological response of 3-BrB-PP1-treated *wee1-as8* cells lacking Cdc42 GEF and GAP activities. Loss of Gef1 did not prevent hyperpolarization of growth at one of the cell tips (Fig. 3a, d). We were unable to assess the role of Scd1 during morphological transition upon Wee1 inhibition, as single *scd1Δ* mutant cells were virtually spherical and did not grow in standard liquid media (Supplementary Fig. 3a), suggesting that Scd1 was essential for normal polarity maintenance in *S. japonicus*.

The lack of the Cdc42 GAP Rga6 produced a noisy response to Wee1 inhibition—some cells polarized normally whereas others failed (Fig. 3b, d, top plot). Strikingly, although *S. japonicus* cells lacking the primary Cdc42 GAP Rga4 were able to maintain cylindrical morphology when grown in normal conditions, they failed to hyperpolarize upon Wee1 inhibition (Fig. 3c, d, top plot). They also failed to reduce cell width in response to 3-BrB-PP1 treatment (Supplementary Fig. 3b). Thus, negative regulators of Cdc42 may spatially constrain its activation to promote hyperpolarized growth.

As *S. japonicus* mutants lacking Rga4 cannot initiate hyperpolarized growth adjusting cellular aspect ratio to a smaller volume (Fig. 3e, f), the 3-BrB-PP1-treated *rga4Δ wee1-as8* cells eventually became virtually spherical, with Pom1 kinase spreading throughout the cellular cortex (Fig. 3g). As both Pom1 and the growth machinery contribute to medial division site positioning in *S. japonicus*[29], Wee1-inhibited *rga4Δ* cells with decreased aspect ratio failed to anchor the actomyosin rings at cell equator, leading to profoundly asymmetric cytokinesis (Fig. 3h, i; see Supplementary Fig. 3c for control time-lapse sequence). When *rga4Δ* cells were shifted from the rich YE to the poorer, chemically defined EMM medium, they were not able to scale their aspect ratio and exhibited severe division site mispositioning resulting in the generation of multinucleated cells (Supplementary Fig. 3d–f, compare with Fig. 1h, i). Taken together, these data indicate that spatial regulation of Cdc42 activity by Rga4 is critical for volume-dependent cellular geometry scaling in *S. japonicus* and contributes to proper patterning of cortical domains.

**Cellular geometry correction rescues positioning errors.** Our data suggested that failure to adapt cellular aspect ratio to cell volume prevents proper division site positioning. We set out to test this hypothesis directly by developing a microfluidics-based experimental setup, where *wee1* mutant *S. japonicus* cells lacking

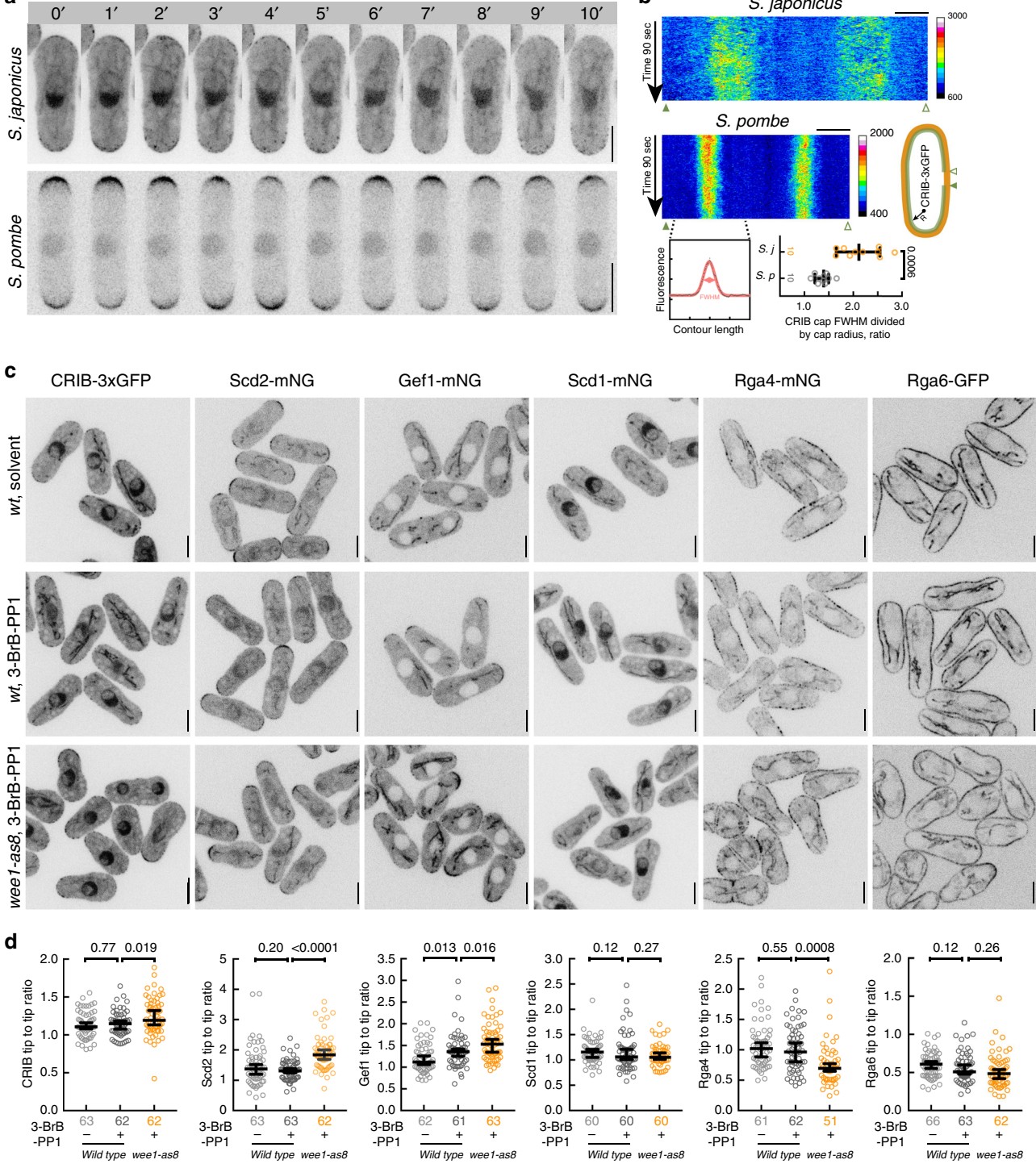

**Fig. 2** Activation zones of the small GTPase Cdc42 rescale during morphological transition. **a** Time-lapse montages of interphase *S. japonicus* (top) and *S. pombe* (bottom) cells expressing a Cdc42 activity reporter CRIB-3xGFP. Shown are maximum intensity *z*-projections of spinning-disk confocal images. **b** Representative kymographs of CRIB-3xGFP at cell periphery in both species (top). 16-Color calibration bars indicate fluorescence intensities in arbitrary units. A diagram showing FWHM fit (full width of the domain at half maximum, bottom left), normalized to cell tip radii (right). **c** Single plane spinning-disk confocal micrographs of cells expressing indicated fluorescent proteins in the wild type (*wt*) and *wee1-as8* genetic backgrounds. Wild type cells were treated either with solvent control or 20 μM 3-BrB-PP1 for 2 h. *wee1-as8* cells were treated with 20 μM 3-BrB-PP1 for 2 h. **d** Graphs show cellular tip-to-tip ratio in fluorescent intensities of markers shown in (**c**). Note that CRIB-3xGFP, Scd2 and Gef1 become enriched at the hyperpolarized cell tip upon Wee1 inhibition, whereas Rga4 is largely excluded from these growing cell protrusions. **a**–**c** Scale bars represent 5 μm. **b**, **d** Quantifications presented as 1D-scatter plots; black bars represent sample median with error bars indicating 95% confidence intervals; *n* indicated in figures; *p* values derived from Kolmogorov–Smirnov test

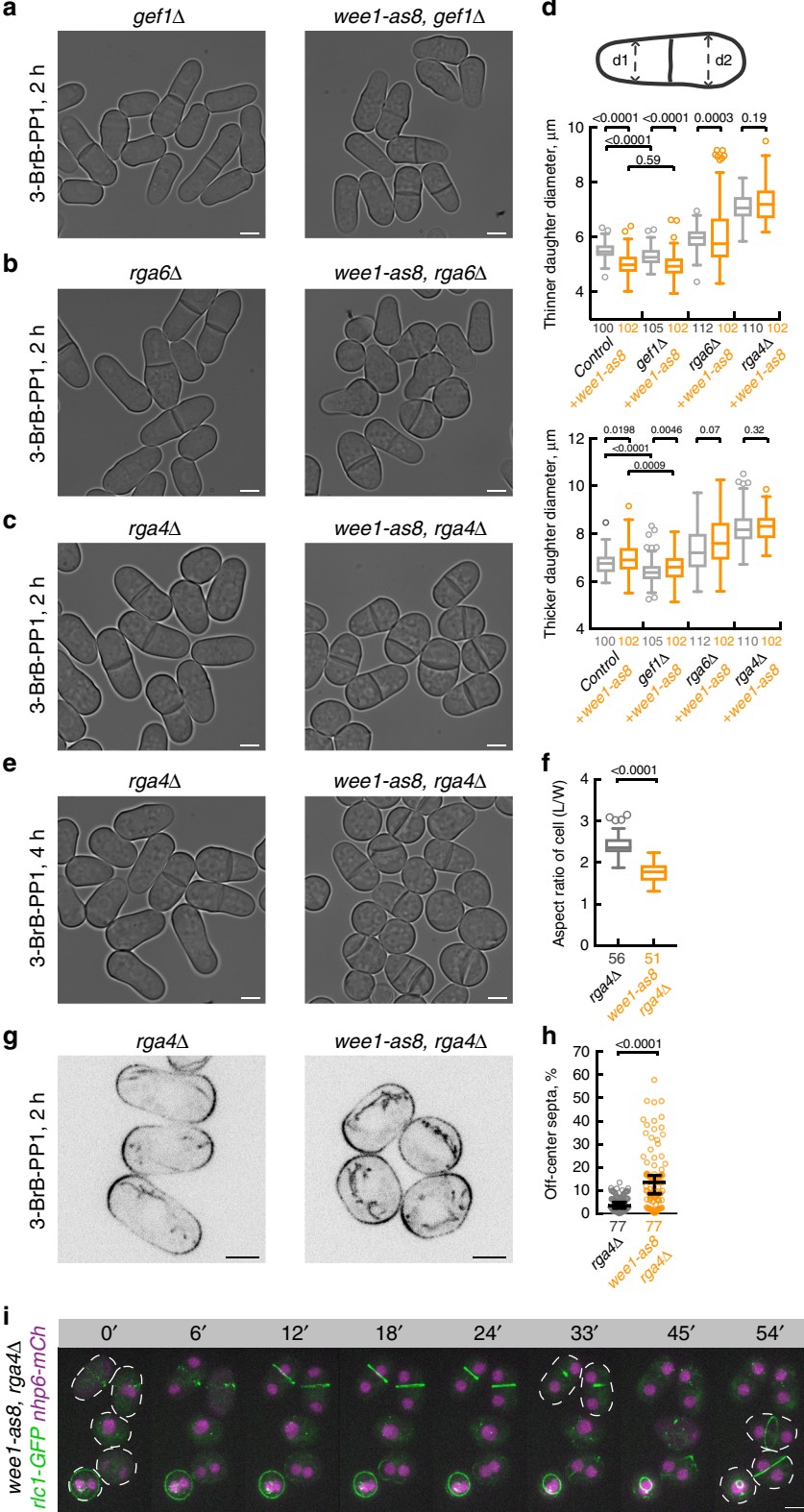

Rga4 could be constrained physically to ensure that they remained cylindrical upon Wee1 inactivation. The overall device design was based on the previously published model[43], but modified to fabricate microchannels of 7 or 10 μm width that could accommodate *S. japonicus* cells. These experiments were performed with cells encoding the temperature-sensitive allele of *wee1*, *wee1-G788E* (Fig. 1f, g), since the non-hydrolysable ATP analogs are known to be absorbed by polydimethylsiloxane (PDMS)[44], used to manufacture microfluidics chambers. The nucleoplasmic protein Nhp6-mCherry was used to assess division site positioning failure resulting in multinucleation. As expected, *wee1-G788E rga4Δ* cells grown in batch cultures were not able to scale upon the temperature shift from 24 °C to 30 °C, and failed in division site positioning (Fig. 4a, d). However, when such a

**Fig. 3** Rga4-dependent rescaling of cellular geometry is essential for medial division plane positioning. **a–c** Single *z*-plane bright-field micrographs of *S. japonicus* cells of indicated genotypes incubated with 20 μM 3-BrB-PP1 (right) for 2 h. Note that Rga4 deficiency prevents hyperpolarization of growth upon Wee1 inhibition. **d** Quantifications of cell diameters in the "thinner" (top) and "thicker" daughter (bottom) cell at cytokinesis, when 3-BrB-PP1 treatment is combined with genetic disruption of Cdc42 module regulators shown in (**a–c**). **e** *wee1-as8 rga4Δ* cells incubated with 20 μM 3-BrB-PP1 (right) for 4 h show severe division site positioning defects. Shown are bright-field micrographs. **f** Quantifications of cellular aspect ratio at division of cells shown in (**e**). **g** Single *z*-plane spinning-disk confocal micrographs of Pom1-GFP-expressing *rga4Δ* and *wee1-as8 rga4Δ* cells treated with 20 μM 3-BrB-PP1 for 2 h. **h** A plot summarizing the accuracy of division plane positioning in cells shown in (**e**). Deviation from the geometric center of the cell is indicated on *y*-axis. Black bars represent sample median. **i** Time-lapse montage of maximum intensity *z*-projected spinning-disk confocal micrographs of 3-BrB-PP1-treated *wee1-as8 rga4Δ S. japonicus* cells expressing Rlc1-GFP and the nucleoplasmic protein Nhp6-mCherry. Cell boundaries are outlined by white dashed lines. Wee1-inhibited *S. japonicus* cells lacking Rga4 fail to anchor the actomyosin ring medially following the semi-open mitosis, as indicated by the dispersal and nuclear re-import of Nhp6. **a–c, e, g, i** Scale bars represent 5 μm. **d, f, h** *n* indicated in figures; *p* values derived using Kolmogorov–Smirnov test

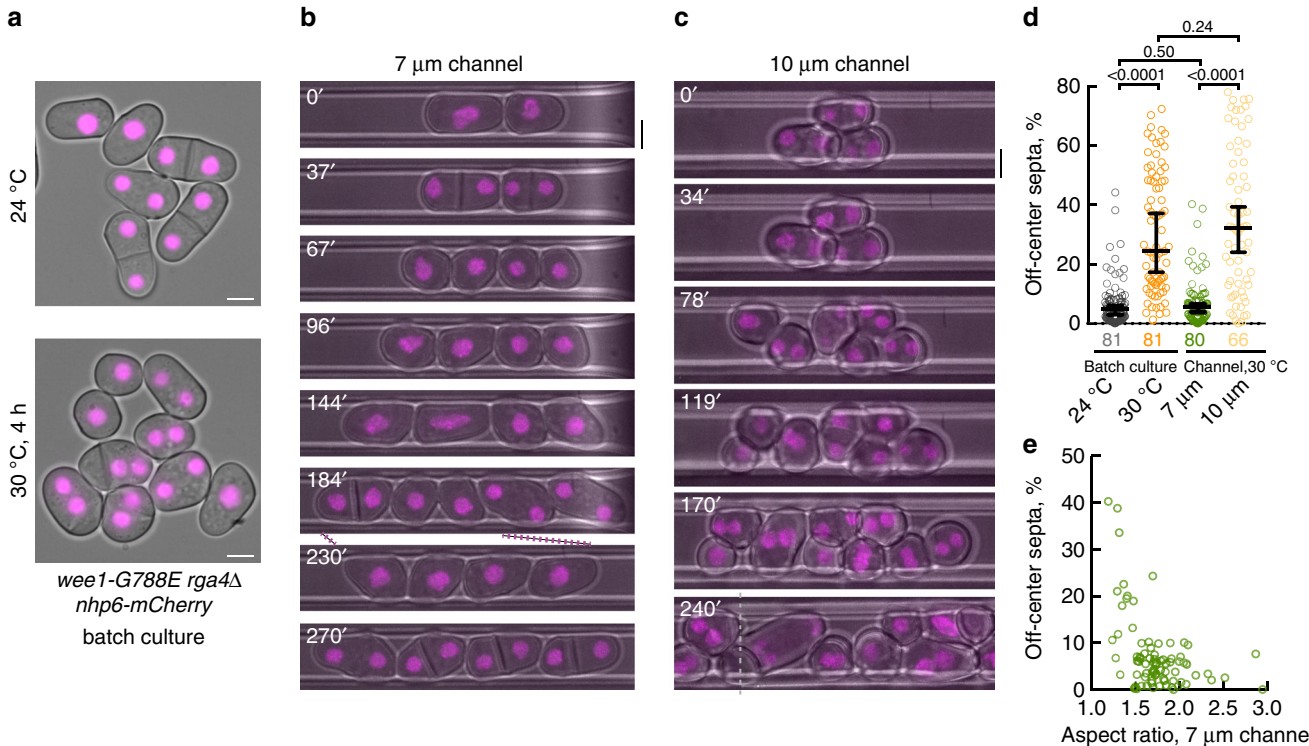

**Fig. 4** Cellular geometry dictates division site positioning in *S. japonicus*. **a** Nhp6-mCherry-expressing *wee1-G788E rga4Δ* cells grown in batch cultures at indicated temperatures. Shown are the pseudocolored epifluorescence images overlaid with bright-field. Note that these cells fail in cytokinesis plane positioning after incubation at 30 °C. **b, c** Nhp6-mCherry-expressing *wee1-G788E rga4Δ* cells grown in 7 μm (**b**) and 10 μm (**c**) channels, respectively, at 30 °C. Shown are the pseudocolored *z*-projected spinning-disk microscope images overlaid with bright-field. Time labels indicate minutes elapsed after cells grown at 24 °C in batch culture were loaded in channels at 30 °C. Gray dotted line in (**c**) indicates the border of two merged images. **d** Graph summarizing the accuracy of division plane positioning of *wee1-G788E rga4Δ* cells after 4-h incubations shown in (**a–c**). **e** A plot showing deviation of division septa from the geometric center of the cell (*y*-axis) vs. cellular aspect ratio (*x*-axis) of individual cells after 4-h incubation at 30 °C in 7 μm channel. **a–c** Scale bars represent 5 μm. Quantifications presented as scatter plots. Black bars represent sample median with error bars indicating 95% confidence intervals; *n* indicated in figures; *p* values derived from Kolmogorov–Smirnov test

temperature shift was performed in the 7 μm microdevice, the daughter cells generated following Wee1 inactivation were forced to grow in a linear pattern. Although the aspect ratio was not fully corrected, most cells remained cylindrical and divided close to cell middle (Fig. 4b, d, e). Quantification of these results suggested that, at least in this system, the minimal length-to-width aspect ratio allowing accurate division site positioning is above 1.5 (Fig. 4b, e). As a control, we used cells grown in broader 10 μm microchannels, which were not able to provide mechanical constraint (Fig. 4c). In this case, most cells failed to remain cylindrical and failed in division site positioning, similarly to batch cultures (Fig. 4d). We concluded that controlling cellular aspect ratio is important for proper division site positioning in *S. japonicus*.

**Aspect ratio control aids cytokinesis positioning in *S. pombe*.** It is broadly accepted in the field that the sister species of *S. japonicus*, *S. pombe*, modulates cell length rather than width when forced to divide at smaller volume. This notion has stemmed from the phenotype exhibited by *wee1* temperature-sensitive mutant cells[30]. Indeed, upon the shift from the permissive temperature of 24 °C to the restrictive temperature of 36 °C, *wee1-50* cells shorten significantly, losing their normal length-to-width aspect ratio (ref. [30] and Fig. 5a, b). Yet, careful examination showed that these mutants in fact became wider (Fig. 5a, b), similarly to the wild type cells that underwent the same treatment (Supplementary Fig. 4a and 4b). We have previously shown that such a temperature shift-up is sufficient to trigger a heat-stress response in *S. pombe*, resulting in transient depolarization of

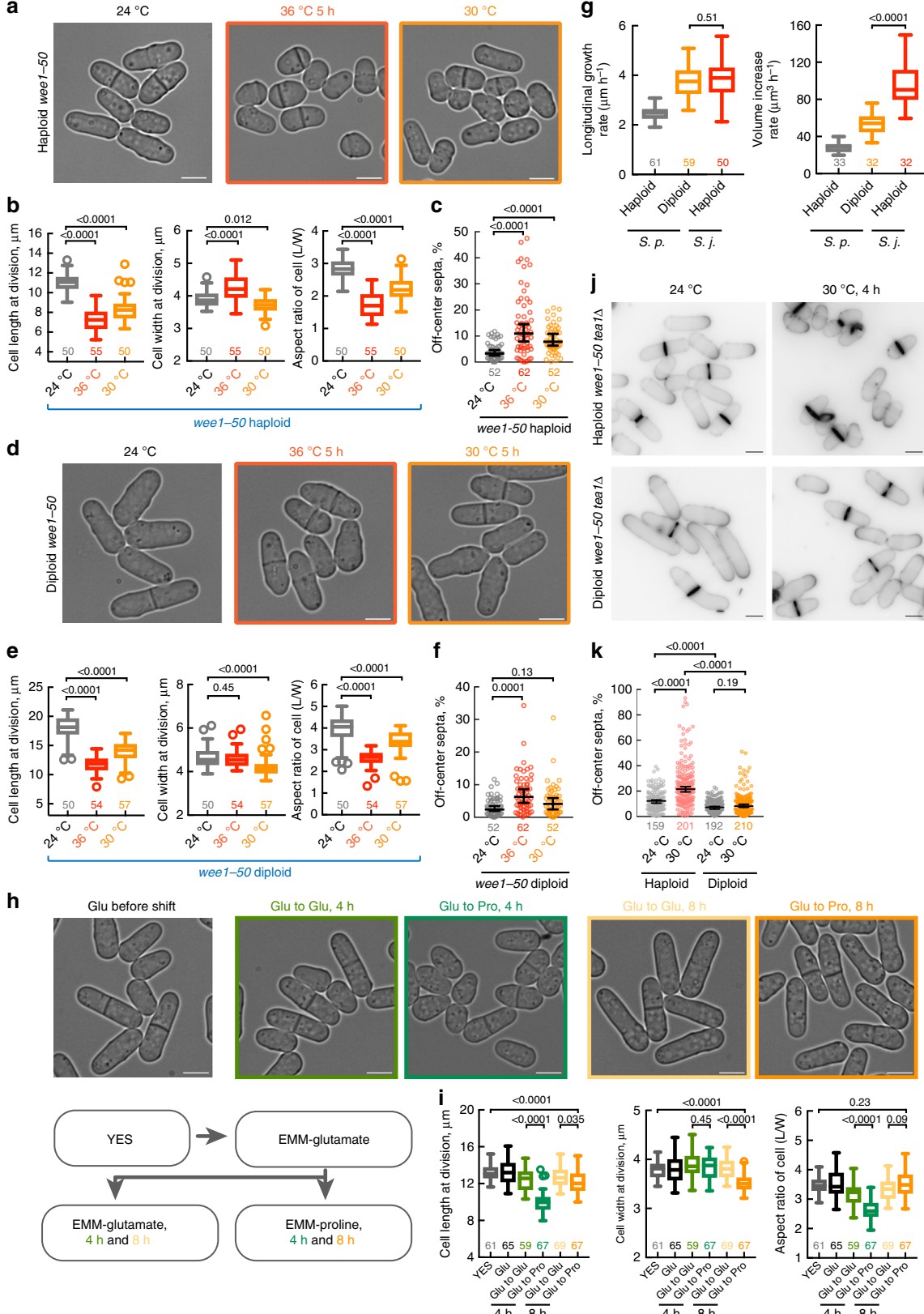

growth and an increase in the cell diameter[45]. In line with cellular polarity defects, *wee1-50* mutants shifted to 36 °C exhibited frequent division site mis-positioning (Fig. 5a, c, see also ref. [34]).

However, *wee1-50* cells exhibited a reduction in cell length at division already at the temperature of 30 °C (Fig. 5a, b).

Interestingly, under these non-stressed conditions, *wee1-50* mutants showed reduced cell diameter, resulting in partial correction of cellular aspect ratio, although the response across the population was quite noisy (Fig. 5a, b). Consistently, we detected occasional division site placement abnormalities

**Fig. 5** Maintaining cellular aspect ratio in *S. pombe* upon reduction in cell volume ensures the fidelity of medial division plane positioning. **a** *S. pombe* temperature-sensitive *wee1-50* haploid cells grown at 24 °C overnight (gray), shifted to 36 °C for 5 h (red) or grown at 30 °C overnight (orange). **b** Quantifications of cell length, width and aspect ratio at division shown in (**a**). **c** A plot summarizing the accuracy of division plane positioning in *wee1-50* *S. pombe* haploid cells grown at indicated temperatures. **d** *S. pombe* *wee1-50* diploid cells grown at 24 °C overnight (gray), shifted to 36 °C for 5 h (red) or to 30 °C for 5 h (orange). **e** Quantifications of cell length, width and aspect ratio shown in (**d**). **f** A plot summarizing the accuracy of division plane positioning in *wee1-50* *S. pombe* diploid cells grown at indicated temperatures. **g** Plots showing the rates of longitudinal growth and cellular volume increase in haploid and diploid *S. pombe* cells, compared with these parameters obtained for haploid *S. japonicus* cells. **h** Wild type *S. pombe* grown overnight in EMM with glutamate as a nitrogen source shifted to glutamate- or proline-based EMM for indicated time. A diagram (bottom) illustrates the workflow of the nitrogen shift experiment. **i** Quantifications of cell length, width and aspect ratio of cell populations represented in (**h**). **j** Calcofluor White-stained live *S. pombe* *wee1-50* *tea1Δ* haploids (top) and diploids (bottom) at indicated temperatures. **k** A plot summarizing the accuracy of division plane positioning of cells shown in (**j**). **a**, **d**, **h** Shown as single *z*-plane bright-field micrographs. Scale bars represent 5 μm. **b**, **e**, **g**, **i** Presented are box-and-whiskers plots using the Tukey method. **c**, **f**, **k** Shown as 1D-scatter plots. Deviation of septa from the geometric center of the cell is indicated on *y*-axis. Black bars represent sample median with error bars indicating 95% confidence intervals. *n* indicated in figures. *p* values derived using Kolmogorov–Smirnov test

although this phenotype was more pronounced in the conditions of heat stress (Fig. 5a, c). This was in contrast to *S. japonicus* *wee1* mutants that were able to scale more efficiently (Fig. 1). Curiously, diploid *wee1-50* *S. pombe* cells exhibited a more coherent scaling of cellular morphology across population after the temperature shift to 30 °C, as compared to haploids (Fig. 5d, e; see Supplementary Fig. 4c, d for wild type diploid data). In line with these data, the diploid *wee1-50* mutant cells exhibited virtually normal division site positioning at 30 °C (Fig. 5d, f). One possible explanation for the variance in scaling efficiency between *S. pombe* haploids and diploids could be higher rates of longitudinal growth and cellular volume increase in diploid cells (Fig. 5g). Arguably, faster polarized growth may allow cells to reach a suitable aspect ratio for optimal division site positioning before the onset of mitosis. In line with this, *S. japonicus*, which is able to scale very efficiently, grows substantially faster as compared to *S. pombe* (Fig. 5g).

We also investigated the behavior of *S. pombe* *wee1-as8* cells[35] upon chemical inhibition of Wee1. The wild type *S. pombe* exhibited hyperpolarization upon incubation with the ATP analog 3-BrB-PP1, consistent with off-target effect on polarity machinery. 3-BrB-PP1-treated cells became slightly thinner and accordingly, increased in length, resulting in higher cellular aspect ratio (Supplementary Fig. 5a, b). This increase did not affect division site positioning (Supplementary Fig. 5c). Unlike in *S. japonicus*, the extent of hyperpolarization in 3-BrB-PP1-treated *S. pombe* *wee1-as8* cells did not differ significantly from the 3-BrB-PP1-treated wild type (compare with Fig. 1). This is perhaps not surprising since both values are quite low in *S. pombe*. However, the Wee1-inhibited cells divided at shorter length and were able to reset cellular aspect ratio after several divisions (Supplementary Fig. 5d, e, 7-h time-point). At earlier time-points of 2 and 4 h post-treatment the populations were considerably heterogeneous, with many cells entering mitosis with abnormal morphology (Supplementary Fig. 5f). Consistently, we observed mild defects in division site placement especially at the 2-h time-point (Supplementary Fig. 5g). Similar to the situation with *wee1-50* mutants, diploid 3-BrB-PP1-treated *wee1-as8* *S. pombe* cells exhibited more coherent and efficient scaling across the population (Supplementary Fig. 5h–j).

To test if scaling was a physiological phenomenon in *S. pombe*, we assessed cellular geometry in prototrophic cells grown in different nitrogen sources[15]. Cells transferred from rich glutamate- to poor proline-based minimal medium initially exhibited pronounced reduction of cell length at division, resulting in lower aspect ratio (Fig. 5h, i, 4-h time-point). Of note, the geometry of these cells eventually recovered, both by reduction in cell width and partial recovery of cell length at division (Fig. 5h, i, 8-h time-point). We concluded that *S. pombe* was able to scale its geometry, similarly to *S. japonicus*.

As shown above, the diploid *S. pombe* cells at steady state already exhibit a higher length-to-width aspect ratio and also are more efficient in geometry scaling as compared to haploids. We wondered if this could facilitate correct positioning of the division site and cell survival under conditions where the "tip inhibition" pathway is compromised and the dominant acto-myosin anchor Mid1 invades cell tips[34]. The tip inhibition pathway depends on the function of the kelch repeat protein Tea1[34]. Consistent with previously published observations[34], we detected severe division site placement abnormalities in haploid Wee1-deficient cells lacking Tea1, which fail to inhibit Mid1-dependent actomyosin ring assembly close to cell tips (Fig. 5j, k; see Supplementary Fig. 5k, l for *wee1-as8* data). This is likely due to the aspect ratios of individual cells transiently dropping below an optimal threshold during morphological transition. As cells enter mitosis with decreased aspect ratio, many will fail in division site positioning resulting in severely decreased population fitness. On the other hand, the diploid *wee1* mutants without Tea1 continued positioning the division site comparatively normally throughout the process of cellular geometry rescaling (Fig. 5j, k; see Supplementary Fig. 5k, l for *wee1-as8* data). Taken together, our results indicate that both *S. pombe* and *S. japonicus* control cellular aspect ratio in proliferating populations. In spite of the two sister species using different strategies to position the division site[23], aspect ratio control is instrumental for the fidelity of cytokinesis in cells dividing at different volumes (Fig. 6).

## Discussion

Cells of many organisms including plants, algae, fungi and bacteria are surrounded by cell walls overlying the plasma membrane. High turgor pressure typical for walled cells provides force to expand the cell. In polarized eukaryotic cells, cellular polarity machinery coordinates membrane trafficking events with cell wall synthesis and remodeling, specifying permissive sites for cell outgrowth. In turn, specific shapes produced by rigid cell wall feedback to the polarity factors, sustaining growth with certain dimensions over generations[41,46]. Yet, most walled cells and multicellular organisms live in complex environments with fluctuating resource availability, necessitating plasticity in cell size determination (for reviews see refs. [47–49]). Strong transgenerational control over morphogenesis may become detrimental when cells encounter different growth conditions.

The fission yeast *S. japonicus*—that normally divides symmetrically –manages the challenge of scaling its shape to lower cell volume by undergoing an asymmetric division. When cell size at birth falls below a certain threshold presumably indicating the history of limited growth in the previous cell cycle, cells hyperpolarize at one of the cell tips, initiating an outgrowth with smaller but stable diameter. This hyperpolarization event follows

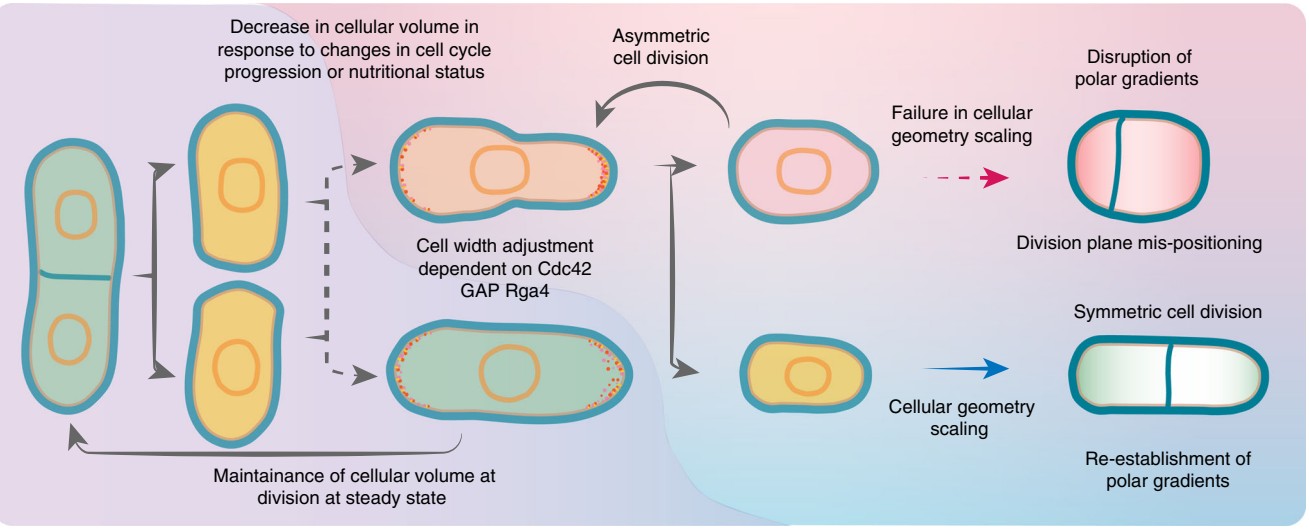

**Fig. 6** Summary diagram. This diagram summarizes our current understanding of cellular geometry rescaling rules following changes in cell volume at division in response to cell cycle or nutritional cues in fission yeasts. Such morphological transition produces two daughter cells with distinct growth modes that eventually influence cortical patterning of the actomyosin for division plane positioning

transient isotropic expansion. Thus, it can be considered conceptually similar to de novo polarization during mating and spore germination in *S. pombe*[50] or bud emergence in budding yeast[51]. Eventually, the entire population returns to symmetric divisions but settles on a new cell size optimum.

At least two aspects of *S. japonicus* biology may contribute to the extent and efficiency of scaling. First, it appears to have a relatively pliable polarity. Although active Cdc42 is enriched at the growing cell tips at levels sufficient to maintain linear growth, it explores wider cortical areas and the degree of tip enrichment at steady state is considerably lower as compared to *S. pombe* (Fig. 2a, b). There could be the significant leeway in tightening this normally relaxed polarity, for instance by manipulating feedback signaling within the Cdc42 polarity module. Second, *S. japonicus* cells (and also diploid *S. pombe*) are much wider to start with and it is possible that there is some minimal diameter determined by intrinsic cellular properties, such as organization of cell wall or polarity machinery, beyond which cells cannot get thinner. Alternatively, the very rapid growth of *S. japonicus* (and again, diploid *S. pombe* cells) could be sufficient for propelling fast tip outgrowth, supporting cellular shape plasticity (Fig. 5g). Both diffuse polarity and rapid growth could be advantageous for a truly dimorphic organism such as *S. japonicus* that alternates between yeast and a rapidly growing hyphal form in the wild[52,53]. Of note, the wild isolates of *S. pombe* differ in growth and polarity traits indicating significant standing variation enabling phenotypic plasticity within the species[54].

The negative regulators of Cdc42, the GAPs Rga6 and in particular, Rga4, are important for scaling in *S. japonicus*, although neither is essential for linear growth at steady state (Fig. 3). It would be of interest to identify the molecular pathway that senses abnormal cellular geometry upon birth—perhaps in conjunction with readout of instantaneous rates of growth—and signals to modify cellular polarity accordingly. Arguably, the efficiency of scaling in *S. japonicus* explains why it can rely solely on the tip inhibition pathway to position the division site[29]. As long as the cellular length-to-width aspect ratio is controlled within population, the pathway is sufficient to produce accurate placement of the division site at the cellular equator. Such readjustment of cellular polarity also becomes important for division site placement in *S. pombe* when cells are forced to divide at smaller volume, especially when one of the actomyosin ring

positioning pathways is compromised (Fig. 5j, k and Supplementary Fig. 5k, l).

Our data suggest that care should be taken in interpreting immediate cellular polarity phenotypes of temperature-sensitive mutants in *S. pombe* and possibly, other ectothermic systems. Even a 12 °C temperature shift-up—staying entirely within the physiological range for this yeast—is sufficient to trigger a transient heat stress response causing cell width increase (ref. [45], Fig. 5a, b and Supplementary Fig. 4). Of note, ATP analog treatments also have clear off-target effects on cellular polarity (Fig. 1b, c and ref. [36]), necessitating multiple lines of evidence to validate experimental conclusions made using these compounds. Although *S. pombe* literature mostly describes cell length at division as a parameter in cell size control[15,35], our results show that cell width can be modified as well, dependent on nutrient conditions or the cell cycle cues (Fig. 5).

Our work identifies *S. japonicus* as a wonderful genetically tractable model system for understanding both the mechanisms and the functional consequences of cellular geometry scaling. It raises a host of exciting questions and puts them within ready experimental reach: How does growth rate contribute to the plasticity of cellular polarity in a walled cell? Does cellular physiology differ depending on cell size? Could a round of asymmetric divisions following changes in growth conditions promote unequal segregation of damaged material or cellular memory determinants within the pair of daughter cells? Finally, it unequivocally establishes that the cellular aspect ratio controls the fidelity of division site positioning in organisms with different approaches to determining the plane of mitotic division.

## Methods

**Yeast strains and growth**. Standard fission yeast media and methods were used[55–58]. Edinburgh minimal medium contained 93.5 mM ammonium chloride, 20 mM glutamate or 20 mM proline as sole nitrogen source. Prototrophic strains of *S. japonicus* and *S. pombe* were used for nutritional switch experiments. See Supplementary Table 1 for a list of *S. japonicus* and *S. pombe* strains. Both yeasts were usually grown overnight at 30 °C in YES (yeast extract with supplements) medium, supplemented with adenine, leucine, histidine and uracil, unless indicated otherwise. Cell cultures with O.D.$_{595\,nm}$ 0.15–0.2 were used as starter cultures for 3-BrB-PP1 treatment or temperature shift experiments. To remove 3-BrB-PP1, cell cultures were centrifuged at 1811×*g*; cell pellet was washed twice and resuspended in fresh YES medium to O.D.$_{595\,nm}$ 0.15–0.2. For nutritional shift experiments, cultured cells (O.D.$_{595\,nm}$ 0.3–0.5) were washed twice and resuspended in the relevant media at O.D.$_{595\,nm}$ 0.15–0.2. Cell densities were kept below O.D.$_{595\,nm}$

0.6 by dilutions, if appropriate. Diploid *S. pombe* cells were obtained by mating of haploid cells of opposite mating types. To prevent sporulation, diploids were maintained on selective solid medium and grown overnight in YES medium. *S. pombe* strains were crossed on YPD solid medium. Mating of *S. japonicus* was performed on SPA solid medium[56]. Spores were dissected and germinated on YES agar plates.

The ATP analog 3-BrB-PP1 (A602985, Toronto Research Chemicals, Canada) was used at a final concentration of 20 μM from a 50 mM stock dissolved in methanol (M/4000, Fisher Chemical). For 3-BrB-PP1 treatments longer than 4 h, additional 3-BrB-PP1 was added at the same concentration. Methanol was used at 1:2500 dilution as solvent control for all 3-BrB-PP1-treatment experiments. Yeast cell walls and division septa were stained with Calcofluor White (1 g L$^{-1}$, 18909, Sigma-Aldrich) at a concentration of 2 mg L$^{-1}$.

**Molecular genetics.** Molecular genetics manipulations were performed using PCR-[59] or plasmid-based homologous recombination[60] using plasmids carrying *S. japonicus* ura4, kanR or natR cloned into the pJK210-backbone (pSO550[27], pSO820 and pSO893, respectively). All primers for cloning and genotyping are shown in Supplementary Table 2. mNeonGreen[61] (under license from Allele Biotechnology; kind gift from K. Gould, Vanderbilt University) was cloned into these plasmids between BamHI and NotI. The *S. japonicus* ATP analog-sensitive wee1-as8 (T582G, M638F amino acid substitutions) and wee1-G788E alleles were generated by fusion PCR using primer sets harboring respective mutations, and integrated at the wee1 locus using pSO550 and pSO820[27]. *S. japonicus* cyclin-dependent kinase cdc2 locus was replaced with the hyperactive allele cdc2-1w (G146D) using pSO550. Construction of the *S. japonicus* CRIB-3xGFP strain was based on ref. [39]; the construct was integrated at the ura4 locus using pSO550[27]. All constructs were sequenced for verification.

**Imaging and data analyses.** Zeiss Axio Observer Z1 fluorescence microscope fitted with α Plan-FLUAR 100×/1.45 NA oil objective lens (Carl Zeiss) and the Orca-Flash4.0 C11440 camera (Hamamatsu) was used to acquire epifluorescence images shown in Figs. 4a and 5j; Supplementary Fig. 3f, Supplementary Fig. 5k and bright-field images shown in Figs. 1, 3–5 and Supplementary Fig. 1-5 unless indicated otherwise. All bright-field images were taken at the medial focal plane of cells unless indicated otherwise.

Spinning-disk confocal images in Figs. 2–4 and Supplementary Figs. 2, 3 were captured by Yokogawa CSU-X1 Spinning Disk Confocal with Eclipse Ti-E Inverted microscope with Nikon CFI Plan Apo Lambda 100× Oil N.A. = 1.45 oil objective, 600 series SS 488 nm, SS 561 nm lasers and Andor iXon Ultra U3-888-BV monochrome EMCCD camera controlled by Andor IQ3. For time-lapse imaging, cells were placed onto agarose pads containing the appropriate growth medium in 2% agarose.

All image analyses were performed using Fiji[62]. Maximum intensity projection of 4–7 z-stacks of 1.0 μm step size images is shown in Fig. 4a, Supplementary Fig. 1a and Supplementary Fig. 3f. Maximum intensity projections of 13–20 z-stacks, 0.5 μm step size images are shown in Figs. 2a (with simple ratio bleach correction for *S. japonicus* CRIB-3xGFP time-lapse), 3i, 4b, c and Supplementary Fig. 3c. 25 z-stacks were acquired with 0.25 μm step size to generate maximum intensity projections shown in Supplementary Fig. 2b-c. All fluorescence images are shown with inverted LUT (look-up table) except in Figs. 2b, 3i, 4a–c, Supplementary Fig. 3c, f.

Kymographs shown in Fig. 2b were constructed by applying the "Multi Kymograph" function to time-lapse single focal plane spinning-disk confocal micrographs of CRIB-3xGFP expressing interphase cells, taken at 2 frames per second for 90 s in total. Image sequence was bleach-corrected using histogram-matching method. Cortical CRIB-3xGFP was traced with segmented line tool. Kymographs were presented as 16-color LUT display of gray-scale image generated using ROI (region of interest) from CRIB cortical tracing with line width of 3 pixels. To estimate the FWHM (full width at half maximum) of CRIB-GFP fluorescence cap at cell ends, we adapted an approach reported in ref. [41]. Briefly, average fluorescent intensity of cortical CRIB-GFP within 25 s was plotted as a function of traced contour length. Least squares fit method was used to fit a Gaussian curve to the plotted function with an estimation of standard deviation (σ) in Prism 7 (GraphPad Software). FWHM was calculated as $2\sqrt{2ln2}\sigma$. Radius of the CRIB fluorescence cap was estimated based on script developed by Olivier Burri, École polytechnique fédérale de Lausanne (EPFL), https://gist.github.com/lacan/42f4abe856f697e664d1062c200fd21f.

To obtain cortical fluorescence profiles of Cdc42 regulators, the contour of cellular cortex of interphase cells was traced with 5-pixel wide line scan. For CRIB-3xGFP, Scd2-NeonGreen, Gef1-NeonGreen and Scd1-NeonGreen the analysis was performed in two ways. To estimate the degree of marker enrichment at one of the cell tips, tip-to-tip ratios were calculated by dividing average intensities of fluorescence markers at the thinner vs. thicker tip (Fig. 2d), Additionally, we calculated "tip enrichment" values, as relative average intensity of fluorescent protein at the cell tip normalized to the overall average intensity of this marker at the cellular cortex (shown in Supplementary Fig. 2d). Note that whereas the latter method reports on marker enrichment, it can be sensitive to cell length. Polarization indexes for Rga4 and Rga6 were calculated by dividing average intensities of these markers at the thinner vs. thicker tip. Fluorescence intensities

of actin along cell periphery of interphase cells (cell length 11–15 μm) was measured and plotted along normalized cell perimeter. Note that cortical localization of Gef1 is particularly sensitive to imaging conditions, and especially to the time cells spend on a slide. Imaging of this marker was typically completed within 5 min.

We assessed cell shape at division using roundness parameter as a proxy, where spherical cells would be assigned a roundness value of 1. Cell roundness values were derived from bright-field images of cells acquired with Zeiss Axio Observer Z1 epifluorescence microscope. Images were corrected for image background and shading using BaSiC plugin[63] available in Fiji/ImageJ. Corrected images were thresholded. Cell contours were outlined using "Analyze Particles" tool and non-cell features were removed during cell segmentation. Cell roundness values were extracted from Fiji build-in measurements of shape descriptors.

Cellular length and width or diameter were measured based on bright-field images acquired with epifluorescence microscope or spinning-disk confocal microscope. To measure cell volume at division in solvent control and 3-BrB-PP1-treated wee1-as8 *S. japonicus* cells, we generated cells expressing mNeonGreen-tagged plasma membrane proton pump Pma1 as a marker for cell boundary[64] and the non-essential myosin regulatory light chain protein Rlc1 fused to mCherry to identify mitotic cells. Volume rendering was performed on Imaris 8.0.0 (Bitplane) based on cell surface created by 0.25 μm step-sized spinning-disk fluorescence images of mNeonGreen-tagged Pma1. Volume measurements in the main text are shown as mean ± standard deviation.

To calculate the volume increase rate and longitudinal growth rate shown in Fig. 5g, pulse-chase staining of cells with FITC-conjugated lectin from *Bandeiraea simplicifolia* was used. Briefly, *S. pombe* haploids and diploids grown at 30 °C in liquid YES medium were incubated with 5 μg ml$^{-1}$ FITC-lectin (L9381, Sigma-Aldrich) for 10 min in the dark, washed twice in YES, resuspended in fresh YES and grown in the dark for up to 150 min before imaging with Zeiss Axio Observer Z1 fluorescence microscope. 20 μg ml$^{-1}$ FITC-lectin was used to stain *S. japonicus* cells due to the weak fluorescence signal obtained at 5 μg ml$^{-1}$ concentration. Medial focal plane of the lectin-stained cells was used to estimate the cell boundary and the lectin-stained zone. Area measurements were obtained in Fiji using manually traced cell boundary and the lectin-stained zone. Growth zones were identified as FITC-lectin free region. Cellular volume increase was estimated using $\Delta V = \pi \cdot (\text{area of cell} - \text{area of lectin-stained zone}) \cdot \text{diameter of cell}/4$, assuming radial symmetry of a cylindrical shape.

**Microfluidic device fabrication and imaging.** Microfluidic channels were fabricated at the Crick Making Lab using soft lithography techniques described in ref. [43]. The overall device design was kindly provided by Nicolas Minc (Institut Jacques Monod, CNRS and Université Paris Diderot), and modified to fabricate microchannels of 7 and 10 μm width, to accommodate *S. japonicus* cells. SU8-2005 and 2007 photoresist (MicroChem) were used to fabricate masters for 7 and 10 μm channels respectively on silicon wafers, Pi-Kem; 3 in., test grade, N (Phos) with Microwriter ML3 (Durham Magneto Optics) at 1 μm resolution. A mixture of Sylgard 184 silicone elastomer and curing agent (Dow; 10:1 ratio w/w) were vacuum-degassed for 1 h at room temperature and polymerized at 110 °C for 15 min. Holes for the inlet and outlet of PDMS chamber were punched using 1.5 mm biopsy puncher. The PDMS side of molded microchannels was bonded to the 25 × 50 mm glass coverslip (Menzel Gläser; thickness: 1.5) after air plasma treatment for 30 s with HPT-100 plasma treater (Henniker Plasma) and incubated at 75°C for 1 h after immediate surface contact.

Cells grown in liquid YES media at 24 °C overnight to OD$_{595 \text{ nm}}$ 0.2 were concentrated 10 times by centrifugation at 1811×g and loaded into the microfluidic device with Plastipak syringe (Becton Dickinson) and tubing (Tygon; ND-100-80, inner diameter: 0.5 mm). Microfluidic device was mounted on the spinning-disk confocal microscope equipped with cage incubator (Okolab) set at 30 °C. Fresh YES liquid media at 30 °C was supplied throughout all microfluidic experiments with flow rates between 20 and 40 μl per minute.

**Quantification and statistical analyses.** Experiments were replicated independently at least twice, and often a larger number of times. Unpaired nonparametric Kolmogorov–Smirnov tests to compare cumulative distributions were performed in Prism 7 (GraphPad Software) to derive p values for all statistical analyses except for Supplementary Fig. 5f and S5j. p values shown in these Supplementary Figures were estimated using Minitab 18 (Minitab Inc.). Box-and-whiskers plots were calculated using the Tukey method in Prism 7. Population medians were indicated in black with error bars representing 95% confidence intervals in all 1D (dimensional) scatter plots. Exact number of cells analyzed was indicated below each column of plots.

**Reporting summary.** Further information on experimental design is available in the Nature Research Reporting Summary linked to this article.

## Data availability
The data that support the findings of this study are available from the corresponding author upon request. The authors declare that all data reported in this study are available within the paper and its Supplementary Information files.

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

## Acknowledgements

The authors are grateful to E. Makeyev for suggestions on the manuscript and M. Balasubramanian, R. Mori and G. Pieper for discussions. Many thanks are due to S. Dundon (Yale University) for communicating the identity of *wee1-50* mutation, I. Hagan (University of Manchester) for *S. pombe wee1-as8* cells, N. Minc (Institute Jacques Monod), D. Coudreuse (IGDR Rennes) and the Crick Making Lab for advice on microfluidics experiments. This work was supported by the Wellcome Trust Senior Investigator Award (103741/Z/14/Z) to S.O.

## Author contributions

Y.G. conceived and performed experiments, analyzed data and wrote the manuscript. S.O. conceived experiments and wrote the manuscript.

## Additional information

**Competing interests:** The authors declare no competing interests.

