## [Peer Review File · Nature Communications]

Reviewers' comments:

Reviewer #1 (Remarks to the Author):

This paper examines how rod-shaped fission yeast cells regulate their geometry upon changes in cell size. The general dogma for *S. pombe* states that these cells maintain a constant width, and adjust cell length as a means to alter cell size. Gu and Oliferenko start by showing that when the fission yeast *S. japonicus* reduces cell size, these cells reduce both their length and width. This maintenance of cell aspect ratio was observed for both genetic mutations and nutrient limitation. The authors studied the mechanism by observing that cells reduce the width of their growth zone in a manner that correlates with activation of Cdc42. Regulation of Cdc42 in these conditions is critical for robust cytokinesis. Finally, they demonstrate that inhibition of *S. pombe* Wee1 causes cells to reduce both their length and width. One reason this observation was missed in decades of previous studies was the difference between temperature-sensitive Wee1 alleles and analog-sensitive Wee1 alleles. The authors show that the two forms of inhibition are not identical because the temperature shift induces a general stress response in addition to Wee1 inhibition.

In general, this is a very interesting paper that questions a long-held dogma in the fission yeast community. More broadly, the paper shows that regulation of cell dimensions has a critical impact of cytokinesis. For this reason, the paper will impact the larger cell biology community. I have a series of mostly minor comments that could be addressed to strengthen the work, in most cases they relate to data presentation and analysis.

Specific comments:

1. Figure S1C-D: This control experiment shows that the inhibitor has some effect on its own. The results should be presented in the same manner as Figure 1D for direct comparison.
2. CRIB marker for activated Cdc42 is described as "rather unstable" in *japonicus*. This is vague, and I don't see evidence for it "exploring a broader area." I would suggest that the authors present the data in a kymograph of a linescan at the cell tips (*japonicus* versus *pombe*). For example, in Figure 2b, I don't even see the colors for the intensity.
3. Figure 2c-e: These results need to be quantified. The authors can measure the % of cells with this localization. It is not clear from the images how many cells show these localization patterns.
4. It would be interesting to know the CRIB-GFP localization in 'skinny' *japonicus* cells, for example grown in minimal media. Do they form a *pombe*-like focused localization pattern for CRIB when growing in sustained hyperpolarized state?
5. In transitioning from *japonicus* to *pombe*, the authors start with the classical *wee1-50* temperature-sensitive mutant. However, this gives confusing results, which ultimately are traced to heat stress. I would suggest starting the *pombe* section with the *wee1-as8* mutant results, which connect more logically with the *japonicus* results. Otherwise, it gets confusing.
6. I did not see simple measurements of wild type *S. pombe* cells treated with the BrBPP1 inhibitor. This control experiment needs to be performed for Figure 4.
7. To broaden the *S. pombe* results, I would suggest that the authors perform their analysis on nutrient conditions that cause reduced cell size at division (i.e. proline as nitrogen source). This experiment would demonstrate if cell width and/or aspect ratio are actively regulated under more physiological conditions than Wee1 inhibition.
8. In my reading the manuscript, the discussion section (pages 13-17) was longer than needed and wandered in places. This section could be condensed to focus on the major conclusions and their implications, and I believe such editing will enhance the paper's appeal to a broad audience.

Reviewer #2 (Remarks to the Author):

In this interesting report, Gu and Oliferenko report that cells of the fission yeast *S. japonicus* preserve their aspect ratio when challenged by inhibition of Wee1 kinase (leading to

hyperactivation of CDK1) or by growth on nutrient-limited medium: these challenges lead to reduction in cell length, but also width. This contrasts with the prevailing view obtained from work using the related *S. pombe*, which mostly adapts its length, a view also partly challenged by results presented here indicating that *S. pombe* also slightly modifies its width. The *S. japonicus* preservation of aspect ratio upon cell length reduction occurs through a first asymmetric division, where one cell end hyperpolarizes and results, upon division, in one thinner daughter cell. The authors further show that hyperpolarization requires the Cdc42 GAP protein Rga4, without which cells are unable to maintain aspect ratio, becoming largely round upon the insults cited above. This in turn leads to cell division positioning errors.

The topic is interesting and brings novel light to an old question, but I have a number of concerns, listed below, that need to be addressed.

Cell thinning phenotype:

I am a bit concerned about how specific the asymmetric division and cell-thinning phenotype is. The authors primarily use an analogue-sensitive *wee1* allele to acutely inhibit it with 3BrB-PP1, but also show that treatment of WT with the inhibitor leads to asymmetric division with one thinner daughter cell (Fig S1c). However, the length, width and aspect ratio of WT cells treated with the drug is not shown. This is problematic. To ensure the phenotype is not an off-target effect of the inhibitor, they then use a *wee1-ts* allele, *wee1-G788E*. The text claims a similar reduction of length and width, suggesting aspect ratio is maintained as in the *wee1-as* allele, but this is not clear to me from the images shown in figure S1e, and the aspect ratio is not shown. These are important missing quantifications.

Similarly, the effect of transfer to low glucose medium is not clear. The text claims similar phenotype as shift to EMM medium, but the images shown in Fig S1g suggest otherwise: cells appear much shorter in length. Quantification of length, width and aspect ratio is required.

In fig 1d, where is the cell width measured? I am a little confused that it looks already as thin after 4h inhibition as after 7h, when panels a-b indicate that cells are still in transition at that time point.

The finding that diploid *S. pombe* cells also maintain aspect ratio upon similar insults is interesting. I find the claimed link between growth rate and aspect ratio maintenance very indirect however. This is at best a loose correlation and there are many other possible reasons for which aspect ratio maintenance may work in diploids and not in haploids. For instance, it could be that there is a minimal width beyond which cells cannot get thinner. This part should be rephrased.

Link to Cdc42 regulation:

The authors claim the cell thinning occurs through hyperpolarization of Cdc42. However, the differences in GEF and GAP localization do not seem obvious, with single highlighted examples not convincing. The whole of fig 2 requires quantification. Regarding the CRIB, it would be helpful to also complement this analysis with examination of the localization of endogenous Bem1/Scd2 homologue as marker of Cdc42-GTP. This would ensure that it is not a question of poor binding of the exogenous CRIB probe.

There are alternative interpretations to the observation that *rga4Δ* cells fail to hyperpolarize in a *wee1* mutant. I would simply interpret that regulation of CDK1 activity is required jointly with Cdc42 GAP for polarized growth. This could be the case if CDK1 activity boosts Cdc42 activity for instance, as is suggested from work in *cerevisiae*, where CDK1 activation of Cdc42 GEF is required

at Start. The shift from YE to EMM may lead to the same effect by some indirect way. One way to test whether the round shape of *rga4Δ wee1*-as upon inhibition represents a failure to hyperpolarize to maintain cellular aspect ratio or a more generic synthetic effect would be to repeat these experiments in haploid *S. pombe*, where aspect ratio is not maintained. If double mutants are also round, this would suggest a more generic synthetic effect.

Along the same lines, the observed hyperpolarization upon *wee1* inhibition may represent a normal situation during the unaltered cell cycle, perhaps at G1/S or early G2, in analogy with Start in budding yeast. Timelapse imaging of Cdc42-GTP may be informative in this respect.

Minor comments:

I am not sure what the authors mean by "bottle-shape" (line 178-179) as opposed to pear-shaped. Does it mean more conical shape? Bottles can be of many different shapes.

Line 190: In what way is Cdc42 "ectopically" activated?

I do not understand the shift in measurement from aspect ratio to "roundness". Why can aspect ratio not be used also for fig 4? It would be easier to compare with the rest of the work.

Reviewer #3 (Remarks to the Author):

The authors explored the phenomenon of cell scaling in *S. japonicus* fission yeast by inducing premature cell division by chemical inhibition of the *wee1* kinase and then measuring the length and width of the resulting cells. The authors found that, as expected, premature division produces cells that are shorter in the axis of growth. They also found, unexpectedly, that these cells grew to be proportionally narrow in the opposite axis, which is normally constant throughout the cell cycle. The remainder of the study focuses on determining the mechanism of this change. The authors imaged various molecules associated with polarity determination in *S. japonicus* and mutated genes in this pathway.

Main concerns are:

1. The authors make the reasonable hypothesis that the *cdc42* pathway is involved in the phenotypes they observe, but none of their experiments demonstrate a direct causal role. Showing that a molecule's localization varies in different organisms does not implicate its function in a phenotype. The authors should employ technologies such as optogenetics to modulate the distribution of active *cdc42* or its regulators and determine the effect on the cell aspect ratio.
2. The only gene mutation that had an effect was *rga4*, but this could be due to its general role in cell polarity, not a specific role in the scaling phenomenon. Combining this and the above point, the data presented in the current manuscript provides little insight on the mechanism of scaling.
3. Although it is nice to show that the *Wee1*-inhibited *rga4* mutant displayed a cytokinetic ring defect, it is unclear whether this was due to the reduced aspect ratio or due to some other abnormalities that result from the loss of *wee1* and *rga4* gene functions. Since this is the only experiment intended to show the physiological importance of the scaling, it needs to be much more convincing. Can the cytokinetic defect be rescued by some independent way of restoring the aspect ratio?
4. The purpose of comparing the two fission yeast species was not well explained. It is unclear if the authors were emphasizing their difference or similarity, or what we can learn from this comparison.

Other specific concerns:

1. The authors did not consistently measure the effect of 3-BrB-PP1 on analog-insensitive cells (i.e. wild-type Wee1 cells). This crucial control for specificity should be included in every experiment where the drug is used. The authors should present this control as a panel in each figure rather than the methanol treatment, which does not control for off-target effects of the drug. This is particularly important where the authors examine active Cdc42. One can imagine treatment with a nucleoside analog could alter GTPase activity directly or indirectly.
2. In Fig. 1d, it seems that cells treated with 3-BrB-PP1 are smaller over the entire observation period. But in Fig1a, it appears that cells are much shorter after 4hr of treatment than 7hr of treatment. Additionally, cells in Fig 1b appear longer than those in the figure 1a after 7hr treatment. This difference is not evident in the box plots.
3. The authors compare the effects of 3-BrB-PP1 treatment with the effect of changing media composition (Fig 1f). Despite phenotypic similarities, this comparison is arbitrary and does not contribute to our understanding of why the cells are narrower.
4. The authors do not show some data that are discussed. All data introduced in an argument should be presented.
5. The authors treated cells with methanol as a control. This was likely used as a solvent for the ATP analog. The authors do not mention how much methanol is added to the cells. This treatment should be referred to as 'vehicle', if appropriate.
6. On page 5, the authors refer to what seems to be a vehicle treatment of analog-sensitive cells as "wild type". This is misleading if the cells were actually wee1-as8 mutants.
7. The authors claim that yeast mutants growing on minimal medium die from division site mis-positioning, yet no experiments address this speculation.
8. The authors include many thoughtful speculations at the end of the manuscript. These are not directly related to the experiments and would be better suited to a perspective article.
9. The authors should structure the manuscript in headed sections for readability.
10. The authors should explain the logic and interpretation behind each experiment without relying so much on the references. For example, the rationale for the use of tea1 mutant in Figure 4 was poorly explained and perhaps only fission yeast experts can understand the purpose of this experiment.
11. Figure legends should be more concise. Many panels are referred to multiple times.

We are very grateful to all reviewers for their insightful comments and suggestions on improving the manuscript. The specific points follow below.

Reviewer 1

This paper examines how rod-shaped fission yeast cells regulate their geometry upon changes in cell size. The general dogma for *S. pombe* states that these cells maintain a constant width, and adjust cell length as a means to alter cell size. Gu and Oliferenko start by showing that when the fission yeast *S. japonicus* reduces cell size, these cells reduce both their length and width. This maintenance of cell aspect ratio was observed for both genetic mutations and nutrient limitation. The authors studied the mechanism by observing that cells reduce the width of their growth zone in a manner that correlates with activation of Cdc42. Regulation of Cdc42 in these conditions is critical for robust cytokinesis. Finally, they demonstrate that inhibition of *S. pombe* Wee1 causes cells to reduce both their length and width. One reason this observation was missed in decades of previous studies was the difference between temperature-sensitive Wee1 alleles and analog-sensitive Wee1 alleles. The authors show that the two forms of inhibition are not identical because the temperature shift induces a general stress response in addition to Wee1 inhibition. In general, this is a very interesting paper that questions a long-held dogma in the fission yeast community. More broadly, the paper shows that regulation of cell dimensions has a critical impact of cytokinesis. For this reason, the paper will impact the larger cell biology community. I have a series of mostly minor comments that could be addressed to strengthen the work, in most cases they relate to data presentation and analysis.

Thank you for your comments. We have added several new pieces of data addressing your specific questions – please see below. We also considerably revised the manuscript according to recommendations by the other two referees.

1. Figure S1C-D: This control experiment shows that the inhibitor has some effect on its own. The results should be presented in the same manner as Figure 1D for direct comparison.

We now present the wild type 3-BrB-PP1 treatment experiment in the same manner as the Wee1 inhibition one (new Supplementary Fig. 1c-e). Briefly, the 3-BrB-PP1 treated wild type *S. japonicus* cells also hyperpolarize although not to the same extent as the *wee1-as* mutant cells ($p < 0.0001$, Kolmogorov-Smirnov tests). As wild type cells become thinner, they elongate so that the overall length-to-width aspect ratio increases. This ~24% aspect ratio increase does not interfere with division site positioning.

2. CRIB marker for activated Cdc42 is described as “rather unstable” in *japonicus*. This is vague, and I don’t see evidence for it “exploring a broader area.” I would suggest that the authors present the data in a kymograph of a linescan at the cell tips (*japonicus* versus *pombe*). For example, in Figure 2b, I don’t even see the colors for the intensity.

Thank you for this suggestion. We have now re-analyzed cortical Cdc42-GTP dynamics. We now show representative linescans of CRIB-GFP at the entire cell periphery in both species (new Fig. 2B, $n=10$), at 2 frames/second time resolution. We hope that you can appreciate that CRIB-GFP recruitment is indeed unstable at the cell tips in *S. japonicus*. To address if this marker explores a broader area around the cell tip in *S. japonicus* (rather than the tip

itself being wider), we have used a method described previously in Bonazzi et al, Current Biology, 2015. In that article, the authors showed that the size of CRIB-GFP domain in *S. pombe* scales with cortical curvature. Here we estimate the full width of the domain at half maximum (FWHM) in both species using Gaussian curve fit and normalize this value to cell tip radii (the procedure is described in Methods). Our results show that these relative FWHM values are significantly different in the two species, with CRIB-GFP in *S. japonicus* exploring a broader area around the cell tips.

3. Figure 2c-e: These results need to be quantified. The authors can measure the % of cells with this localization. It is not clear from the images how many cells show these localization patterns.

We have quantified all experiments shown in Fig. 2 and provide the statistics adjacent to the representative images. To address the point made by the Reviewer 3, we now show the wild type *S. japonicus* cells expressing fluorescent Cdc42 regulators/markers treated with 3-BrB-PP1 as controls. Additionally, we now include new data for another diagnostic marker of Cdc42 activity, Scd2-NeonGreen (new Fig. 2d), as requested by the Reviewer 2. For markers that – at least from *S. pombe* work – are expected to accumulate at growing cell tips (CRIB-GFP, Scd2-NeonGreen, Gef1-NeonGreen and Scd1-NeonGreen), we obtained ‘tip enrichment’ values by measuring average intensity of the relevant fluorescent protein at the cell tip and normalizing it to the overall average intensity of this marker at the cellular cortex. As can be appreciated from these measurements, both markers of Cdc42 activity, CRIB-GFP (new Fig. 2c) and Scd2-NeonGreen (new Fig. 2d), are enriched at the thinner cell tip in scaling *S. japonicus* cells. Gef1 also shows a similar enrichment at hyperpolarizing tips in *wee1-as8* mutant cells. Although the second GEF, Scd1, is virtually essential for polarized growth in *S. japonicus* (Supplementary Fig. 3a), we do not observe a statistically significant enrichment of this protein at the growing tips neither in wild type (untreated or 3-BrB-PP1 treated) nor in thinning *wee1-as* mutant cells, suggesting that this regulator does not have to be enriched at specific sites to exert its function in *S. japonicus* (new Fig. 2f). In case of GAPs that are often excluded from the growing tips, we used a different type of statistics. Even in untreated wild type *S. japonicus*, one of the cell tips – often the more growing one – tends to be slightly pointier. We calculated polarization indexes for Rga4 and Rga6 by dividing average intensities of these markers at the thinner vs thicker tip. As can be seen from our data, Rga4-NeonGreen shows significant clearing from the thinning tip during the process of scaling (Fig. 2g). We did not observe significant differences in tip clearing in the case of the other GAP, Rga6, as compared to 3-BrB-PP1 treated wild type cells (Fig. 2h). In general, Rga6 in *S. japonicus* exhibits a much broader distribution along the cellular cortex, including cell tips, as compared to Rga4, especially in shorter post-cytokinetic cells.

4. It would be interesting to know the CRIB-GFP localization in ‘skinny’ *japonicus* cells, for example grown in minimal media. Do they form a pombe-like focused localization pattern for CRIB when growing in sustained hyperpolarized state?

Yes, we were very curious about this! It turns out that the CRIB-GFP localization either in *wee1-as* cells grown for longer time in the presence of 3-BrB-PP1 or the wild type cells grown in minimal medium overnight goes back to a 'normal' situation, that is, very weak enrichment at the cell tips. In fact, it is even more depleted in cells grown in EMM. We now provide representative images in Supplementary Fig. 2a.

5. In transitioning from *japonicus* to *pombe*, the authors start with the classical *wee1-50* temperature-sensitive mutant. However, this gives confusing results, which ultimately are traced to heat stress. I would suggest starting the *pombe* section with the *wee1-as8* mutant results, which connect more logically with the *japonicus* results. Otherwise, it gets confusing.

During this revision, we have put a lot of effort into thorough investigation of physiological effects produced by 3-BrB-PP1 in *S. pombe*. Our conclusion is that this inhibitor produces mild hyperpolarization in the wild type background (Supplementary Fig. 5a-b). Since wild type cells maintain cell volume at division, they become longer, resulting in an increased aspect ratio (Supplementary Fig. 5a-b). Such an increase is not translated into division site positioning problems (Supplementary Fig. 5c). Wee1-analog-inhibited cells show quite a different phenotype – they become thinner and shorter. Having said that, the absolute difference in the extent of hyperpolarization is not significantly different between 3-BrB-PP1-treated wild type and *wee1-as* cells. This is perhaps not surprising since both of these values are pretty low in haploid *S. pombe* cells. Of note, the difference is much more obvious in *S. japonicus* cells, which are better at scaling. To make a clear point on scaling in *S. pombe*, we have thus decided to present our *wee1-50* temperature-sensitive mutant data in the main body of the manuscript (new Fig. 5). We have extended this piece of work to cover the division site positioning abnormalities in Tea1-deficient cells (new Fig. 5j-k).

Our main conclusions remain unchanged. First, we show that haploid *S. pombe* cells can undergo some geometrical scaling when forced to divide at smaller volume. Second, we demonstrate that diploid *S. pombe* cells are able to do it more effectively. Third, we show that the efficiency of geometric scaling has a bearing on the accuracy of division site positioning in *S. pombe* cells.

We think it is a good idea to show our results on 3-BrB-PP1-treatment in Supplementary Data. First, it is important for the field to be aware of the phenotypes exerted by this widely used inhibitor the wild type cells. Second, Wee1 inhibition using 3-BrB-PP1 does produce geometrical scaling (new Supplementary Fig. 5), with clear consequences for the fidelity of division site positioning, thus corroborating our experiments with *wee1-50*.

6. I did not see simple measurements of wild type *S. pombe* cells treated with the BrBPP1 inhibitor. This control experiment needs to be performed for Figure 4.

Please see above.

7. To broaden the *S. pombe* results, I would suggest that the authors perform their

analysis on nutrient conditions that cause reduced cell size at division (i.e. proline as nitrogen source). This experiment would demonstrate if cell width and/or aspect ratio are actively regulated under more physiological conditions that Wee1 inhibition.

Thank you for prompting us to do this experiment. We have compared cellular geometry in *S. pombe* cells grown in different nitrogen sources (glutamate vs proline). Indeed, cells grown in proline as a sole nitrogen source become thinner and control aspect ratio (new Fig. 5h-i).

8. In my reading the manuscript, the discussion section (pages 13-17) was longer than needed and wandered in places. This section could be condensed to focus on the major conclusions and their implications, and I believe such editing will enhance the paper's appeal to a broad audience.

Agreed, we shortened the Discussion. Hopefully you will find the new version more to the point.

Reviewer 2

In this interesting report, Gu and Oliferenko report that cells of the fission yeast *S. japonicus* preserve their aspect ratio when challenged by inhibition of Wee1 kinase (leading to hyperactivation of CDK1) or by growth on nutrient-limited medium: these challenges lead to reduction in cell length, but also width. This contrasts with the prevailing view obtained from work using the related *S. pombe*, which mostly adapts its length, a view also partly challenged by results presented here indicating that *S. pombe* also slightly modifies its width. The *S. japonicus* preservation of aspect ratio upon cell length reduction occurs through a first asymmetric division, where one cell end hyperpolarizes and results, upon division, in one thinner daughter cell. The authors further show that hyperpolarization requires the Cdc42 GAP protein Rga4, without which cells are unable to maintain aspect ratio, becoming largely round upon the insults cited above. This in turn leads to cell division positioning errors.

The topic is interesting and brings novel light to an old question, but I have a number of concerns, listed below, that need to be addressed.

Thank you for your comments – please see our detailed responses to your concerns and suggestions below.

Cell thinning phenotype:

I am a bit concerned about how specific the asymmetric division and cell-thinning phenotype is. The authors primarily use an analogue-sensitive *wee1* allele to acutely inhibit it with 3BrB-PP1, but also show that treatment of WT with the inhibitor leads to asymmetric division with one thinner daughter cell (Fig S1c). However, the length, width and aspect ratio of WT cells treated with the drug is not shown. This is problematic. To ensure the phenotype is not an off-target effect of the inhibitor, they then use a *wee1-ts* allele, *wee1-G788E*. The text claims a similar reduction of length and width, suggesting aspect ratio is maintained as in the *wee1-as* allele, but this is not clear to me from the images shown in figure S1e, and the aspect ratio is not shown. These are important missing quantifications.

We now present all relevant quantifications of the phenotype produced by treatment of wild type *S. japonicus* cells with 3-BrB-PP1 (new Supplementary Fig. 1c-e). Briefly, the 3-BrB-PP1 treated wild type *S. japonicus* cells hyperpolarize although not to the same extent as the *wee1-as* mutant cells

($p < 0.0001$, Kolmogorov-Smirnov tests). As wild type cells become thinner, they elongate so that the overall length-to-width aspect ratio increases. This ~24% aspect ratio increase does not interfere with division site positioning.

Similarly, the effect of transfer to low glucose medium is not clear. The text claims similar phenotype as shift to EMM medium, but the images shown in Fig S1g suggest otherwise: cells appear much shorter in length. Quantification of length, width and aspect ratio is required.

This is a tricky experiment to quantitate. We do not have a chemostat and we were not able to get low glucose experiments working reliably in microfluidics setups because cell width changes and cells are not retained under flow. Instead, we performed them in bulk cultures. The issue with this is that at some point cells do run out of glucose and stop growing. This results in a fairly noisy phenotype in overnight cultures where most cells do get thinner but the length at division is highly variable. We have refined this experiment and we now obtain much more coherent datasets from 0.2% glucose concentration (presented in Supplementary Fig. 1j-k). Since this is not a major point of the paper, we hope that you will agree that these results would be sufficient.

In fig 1d, where is the cell width measured? I am a little confused that it looks already as thin after 4h inhibition as after 7h, when panels a-b indicate that cells are still in transition at that time point.

We measure the cell width at septum. At 4 hours, most asymmetric cells divide close to the neck, hence the septum spans a thinner half of the cell.

The finding that diploid *S. pombe* cells also maintain aspect ratio upon similar insults is interesting. I find the claimed link between growth rate and aspect ratio maintenance very indirect however. This is at best a loose correlation and there are many other possible reasons for which aspect ratio maintenance may work in diploids and not in haploids. For instance, it could be that there is a minimal width beyond which cells cannot get thinner. This part should be rephrased.

Point taken, we rephrased this part, highlighting this possibility in Discussion.

Link to Cdc42 regulation:

The authors claim the cell thinning occurs through hyperpolarization of Cdc42. However, the differences in GEF and GAP localization do not seem obvious, with single highlighted examples not convincing. The whole of fig 2 requires quantification. Regarding the CRIB, it would be helpful to also complement this analysis with examination of the localization of endogenous Bem1/Scd2 homologue as marker of Cdc42-GTP. This would ensure that it is not a question of poor binding of the exogenous CRIB probe.

This is the point also made by the Reviewer 1. We copy the relevant bit of our response below.

We have quantified all experiments shown in Fig. 2 and provide the statistics adjacent to the representative images. To address the point made by the Reviewer 3, we now show the wild type *S. japonicus* cells expressing fluorescent Cdc42 regulators/markers treated with 3-BrB-PP1 as controls. Additionally, we now include new data for another diagnostic marker of Cdc42

activity, Scd2-NeonGreen (new Fig. 2d), as requested by the Reviewer 2. For markers that – at least from *S. pombe* work – are expected to accumulate at growing cell tips (CRIB-GFP, Scd2-NeonGreen, Gef1-NeonGreen and Scd1-NeonGreen), we obtained ‘tip enrichment’ values by measuring average intensity of the relevant fluorescent protein at the cell tip and normalizing it to the overall average intensity of this marker at the cellular cortex. As can be appreciated from these measurements, both markers of Cdc42 activity, CRIB-GFP (new Fig. 2c) and Scd2-NeonGreen (new Fig. 2d), are enriched at the thinner cell tip in scaling *S. japonicus* cells. Gef1 also shows a similar enrichment at hyperpolarizing tips in *wee1-as8* mutant cells. Although the second GEF, Scd1, is virtually essential for polarized growth in *S. japonicus* (Supplementary Fig. 3a), we do not observe a statistically significant enrichment of this protein at the growing tips neither in wild type (untreated or 3-BrB-PP1 treated) nor in thinning *wee1-as* mutant cells, suggesting that this regulator does not have to be enriched at specific sites to exert its function in *S. japonicus* (new Fig. 2f). In case of GAPs that are often excluded from the growing tips, we used a different type of statistics. Even in untreated wild type *S. japonicus*, one of the cell tips – often the more growing one – tends to be slightly pointier. We calculated polarization indexes for Rga4 and Rga6 by dividing average intensities of these markers at the thinner vs thicker tip. As can be seen from our data, Rga4-NeonGreen shows significant clearing from the thinning tip during the process of scaling (Fig. 2g). We did not observe significant differences in tip clearing in the case of the other GAP, Rga6, as compared to 3-BrB-PP1 treated wild type cells (Fig. 2h). In general, Rga6 in *S. japonicus* exhibits a much broader distribution along the cellular cortex, including cell tips, as compared to Rga4, especially in shorter post-cytokinetic cells.

There are alternative interpretations to the observation that *rga4Δ* cells fail to hyperpolarize in a *wee1* mutant. I would simply interpret that regulation of CDK1 activity is required jointly with Cdc42 GAP for polarized growth. This could be the case if CDK1 activity boosts Cdc42 activity for instance, as is suggested from work in cerevisiae, where CDK1 activation of Cdc42 GEF is required at Start. The shift from YE to EMM may lead to the same effect by some indirect way. One way to test whether the round shape of *rga4Δ wee1-as* upon inhibition represents a failure to hyperpolarize to maintain cellular aspect ratio or a more generic synthetic effect would be to repeat these experiments in haploid *S. pombe*, where aspect ratio is not maintained. If double mutants are also round, this would suggest a more generic synthetic effect.

We performed this experiment. Haploid *wee1-as rga4Δ S. pombe* cells do not lose polarity upon treatment with 3-BrB-PP1 (please see the figure below).

We believe this is in line with *S. pombe* being considerably more ‘polarized’, with several pathways reinforcing linear growth pattern. In our experience, a number of insults that virtually depolarize the *S. japonicus* cell (e.g. *for3* Δ , *scd1* Δ), are not sufficient to produce such strong phenotypes in *S. pombe*.

To address if it is indeed the cellular aspect ratio that ensures the fidelity of division site positioning in *S. japonicus*, we have developed a microfluidics based setup, where *wee1-ts rga4* Δ *S. japonicus* cells can be constrained to prevent complete depolarization upon shift to 30°C. We now provide an entirely new Fig. 4 that describes this set of experiments. Briefly, we show that constraining cell diameter and hence, at least partially rescuing cell length-to-width aspect ratio, leads to profound rescue of division site positioning in cells dividing at smaller volume and in the absence of Rga4. Please note, that we performed these experiments with the temperature-sensitive allele of *wee1*, rather than with an analog-sensitive one, since ATP analogs, just like many other chemicals, are absorbed by PDMS that we must use to make very fine channels allowing cells to be constrained (see a recent paper from the Coudreuse lab (PMID 27512142) that addresses PDMS-related problems in detail).

Along the same lines, the observed hyperpolarization upon *wee1* inhibition may represent a normal situation during the unaltered cell cycle, perhaps at G1/S or early G2, in analogy with Start in budding yeast. Timelapse imaging of Cdc42-GTP may be informative in this respect.

We did look at CRIB-GFP and Scd2-NeonGreen dynamics during the normal cell cycle – although not at a very high time resolution to prevent bleaching and photo-damage during long period of imaging. Cell cycle timing in *S. japonicus* growing in rich medium is similar to *S. pombe*, with very short G1, fast S and long G2 phases (our unpublished data that were obtained using DNA FACS and Tos4 and Rpa1 as progression markers). Cell separation occurs largely in early G2. In imaging experiments, we do not observe any specific time point in the cell cycle at which cells hyperpolarize similarly to

Wee1-deficient cells, meaning changes in cell width. Cdc42 activity markers tend to be slightly enriched at the growing tips immediately after septation but it is not translated into persistent occupancy and hyperpolarized growth. We fully agree that it is a great idea to investigate the relationship between the cell cycle and polarized growth, and we will do it properly, but we believe it is beyond the scope of this manuscript. A typical montage of a time-lapse is shown below.

Minor comments:

I am not sure what the authors mean by "bottle-shape" (line 178-179) as opposed to pear-shaped. Does it mean more conical shape? Bottles can be of many different shapes.

We edited the text.

Line 190: In what way is Cdc42 "ectopically" activated?

Rephrased.

I do not understand the shift in measurement from aspect ratio to "roundness". Why can aspect ratio not be used also for fig 4? It would be easier to compare with the rest of the work.

We used 'roundness' (broadly, an inverse of aspect ratio) in this instance because we believed that it was a more intuitive way to show the noisy response of haploid population during transition. That said, we now changed the order of figures, showing *S. pombe wee1-50* results in primary Fig. 5, where we only show aspect ratio values. Wee1-as results are now shown in Supplemental Fig. 5.

Reviewer 3

The authors explored the phenomenon of cell scaling in *S. japonicus* fission yeast by inducing premature cell division by chemical inhibition of the *wee1* kinase and then measuring the length and width of the resulting cells. The authors found that, as expected, premature division produces cells that are shorter in the axis of growth. They also found, unexpectedly, that these cells grew to be proportionally narrow in the opposite axis, which is normally constant throughout the cell cycle. The remainder of the study focuses on determining the mechanism of this change. The authors imaged various molecules associated with polarity determination in *S. japonicus* and mutated genes in this pathway. Main concerns are:

1. The authors make the reasonable hypothesis that the *cdc42* pathway is involved in the phenotypes they observe, but none of their experiments demonstrate a direct causal role. Showing that a molecule's localization varies in different organisms does not implicate its function in a phenotype. The authors should employ technologies such as optogenetics to modulate the distribution of active *cdc42* or its regulators and determine the effect on the cell aspect ratio.

Thank you for your comments. It would indeed be very useful to use optogenetics to probe directly the roles of Cdc42 module in polarity establishment and maintenance during hyperpolarization and normal growth. However, such a setup has not been published even for *S. pombe*, a widely studied model organism, and to our knowledge, does not exist for *S. japonicus*. Our understanding from the field is that a number of labs have put a lot of efforts into designing the working system in *S. pombe* and we will hopefully see an optimized system at some point in future and will be able to translate it to *S. japonicus*. It has been a difficult problem to solve, as compared to budding yeast.

Regarding the role of Cdc42 polarity module in scaling, the main point we have been making is that the Cdc42 GAP Rga4 is required for this process. In this manuscript, we do not attempt to prove the position of Cdc42 in the pathway that triggers and sustains hyperpolarized growth. Importantly, we provide a first description of cell polarity machinery in the new model organism.

2. The only gene mutation that had an effect was *rga4*, but this could be due to its general role in cell polarity, not a specific role in the scaling phenomenon. Combining this and the above point, the data presented in the current manuscript provides little insight on the mechanism of scaling.

Rga4 is not required for polarized growth per se and whereas *rga4Δ S. japonicus* and *S. pombe* cells are slightly wider, they continue to grow in a linear pattern (e.g. Fig. 3c). The defect we see is related specifically to scaling. It is also worth mentioning that the other GAP, Rga6, also has a mild phenotype in scaling, suggesting that negative regulation of Cdc42 activity plays an important part in this process.

3. Although it is nice to show that the Wee1-inhibited *rga4* mutant displayed a cytokinetic ring defect, it is unclear whether this was due to the reduced aspect ratio or due to some other abnormalities that result from the loss of *wee1* and *rga4* gene functions. Since this is the only experiment intended to show the physiological importance of the scaling, it needs to be much more convincing. Can the cytokinetic defect be rescued by some independent way of restoring the aspect ratio?

To address your point, we have developed a microfluidics based setup, where *wee1-ts rga4Δ S. japonicus* cells can be constrained to prevent complete depolarization upon shift to 30°C. We now provide an entirely new Fig. 4 that describes this set of experiments. Briefly, we show that constraining cell diameter and hence, at least partially rescuing cell length-to-width aspect ratio, leads to profound rescue of division site positioning in cells dividing at smaller volume and in the absence of Rga4. Please note, that we performed these experiments with the temperature-sensitive allele of *wee1*, rather than

with an analog-sensitive one, since ATP analogs, just like many other chemicals, are absorbed by PDMS that we must use to make very fine channels allowing cells to be constrained (see a recent paper from the Coudreuse lab (PMID 27512142) that addresses PDMS-related problems in detail).

We hope that you will find this set of data supporting our statement that cellular length-to-width aspect ratio ensures the fidelity of division site positioning convincing.

4. The purpose of comparing the two fission yeast species was not well explained. It is unclear if the authors were emphasizing their difference or similarity, or what we can learn from this comparison.

We made an effort of explaining this better in the text.

Other specific concerns:

1. The authors did not consistently measure the effect of 3-BrB-PP1 on analog-insensitive cells (i.e. wild-type Wee1 cells). This crucial control for specificity should be included in every experiment where the drug is used. The authors should present this control as a panel in each figure rather than the methanol treatment, which does not control for off-target effects of the drug. This is particularly important where the authors examine active Cdc42. One can imagine treatment with a nucleoside analog could alter GTPase activity directly or indirectly.

Thank you for this suggestion. We took it very seriously and investigated the effect of 3-BrB-PP1 on both wild type *S. pombe* and *S. japonicus*. Our controls are now shown as requested.

Our conclusion is that this inhibitor produces mild hyperpolarization in wild type cells of both species (Supplementary Fig. 1c-e and Supplementary Fig. 5a-b). Since wild type cells maintain cell volume at division, they become longer, resulting in an increased aspect ratio. Such an increase is not translated into division site positioning problems. In the case of *S. japonicus* that scales very well, the extent of hyperpolarization in *wee1-as* mutant cells is significantly larger ($p < 0.0001$, Kolmogorov-Smirnov test). In the case of *S. pombe*, the absolute difference in the extent of hyperpolarization is not significantly different between 3-BrB-PP1-treated wild type and *wee1-as* cells. This is perhaps not surprising since both of these values are already low in haploid *S. pombe* cells. To make a clear point on scaling in *S. pombe*, we have thus decided to present our *wee1-50* temperature-sensitive mutant data in the main body of the manuscript (new Fig. 5). We have extended this piece of work to cover the division site positioning abnormalities in Tea1-deficient cells (new Fig. 5j-k).

Our main conclusions remain unchanged. First, we show that haploid *S. pombe* cells can undergo some geometrical scaling when forced to divide at smaller volume. Second, we demonstrate that diploid *S. pombe* cells are able to do it more effectively. Third, we show that the efficiency of geometric scaling has a bearing on the accuracy of division site positioning in *S. pombe* cells.

2. In Fig. 1d, it seems that cells treated with 3-BrB-PP1 are smaller over the entire observation period. But in Fig1a, it appears that cells are much shorter after 4hr of treatment than 7hr of treatment. Additionally, cells in Fig 1b appear longer than those in the figure 1a after 7hr treatment. This difference is not evident in the box plots.

A field of cells shown in Fig. 1a (e.g. 4h time point) include both interphase cells and cells that undergoing cytokinesis (with division septa). Our quantifications of cell length, width and the length-to-width aspect ratio (Fig. 1c) relate to septated cells specifically, since we discuss these values at division.

Related to the second point, experiments shown in Fig. 1a and Fig. 1d (these are the new labels) are separate since the entire cycle of 'scaling' and 're-scaling' is too long to fit into one working day. Shown are the representative examples of these experiments. Cells in Fig. 1d are indeed a bit longer which can be also seen in quantifications (Fig. 1e). This does not affect our conclusions that *S. japonicus* cells can regain their normal size and geometry upon return to the normal cell cycle.

3. The authors compare the effects of 3-BrB-PP1 treatment with the effect of changing media composition (Fig 1f). Despite phenotypic similarities, this comparison is arbitrary and does not contribute to our understanding of why the cells are narrower.

We believe that it is very important to provide evidence that cells maintain their geometry in different physiological conditions when they divide at smaller volume. Whereas the exact molecular pathway that leads to scaling in these two cases could indeed be different, the end result is the same, i.e. the lack of scaling prevents cells to position the division site accurately.

4. The authors do not show some data that are discussed. All data introduced in an argument should be presented.

Point taken, we edited the manuscript accordingly.

5. The authors treated cells with methanol as a control. This was likely used as a solvent for the ATP analog. The authors do not mention how much methanol is added to the cells. This treatment should be referred to as 'vehicle', if appropriate.

We now mention these details in Methods. We refer to methanol as solvent control.

6. On page 5, the authors refer to what seems to be a vehicle treatment of analog-sensitive cells as "wild type". This is misleading if the cells were actually *wee1-as8* mutants.

Sorry about it, corrected!

7. The authors claim that yeast mutants growing on minimal medium die from division site mis-positioning, yet no experiments address this speculation.

We now show severe bi-nucleation and division site mispositioning in *rga4Δ* *S. japonicus* cells transferred to the minimal medium (Supplementary Fig. 3e).

8. The authors include many thoughtful speculations at the end of the manuscript. These are not directly related to the experiments and would be better suited to a perspective article.

Point taken, we edited the manuscript accordingly.

9. The authors should structure the manuscript in headed sections for readability.

Thank you for this suggestion. We edited the manuscript accordingly.

10. The authors should explain the logic and interpretation behind each experiment without relying so much on the references. For example, the rationale for the use of tea1 mutant in Figure 4 was poorly explained and perhaps only fission yeast experts can understand the purpose of this experiment.

Thank you for this feedback. We provided clarification in this instance and, in general, to improve readability.

11. Figure legends should be more concise. Many panels are referred to multiple times.

We shortened the Figure legends.

REVIEWERS' COMMENTS:

Reviewer #1 (Remarks to the Author):

The authors have addressed my comments and have improved the manuscript nicely. This paper has a number of interesting results and conclusions that are likely to appeal to a broad cell biology audience.

My one lingering concern is the effect of 3-BrB-PP1 on cell width. The authors performed the requested control experiment by testing how 3-BrB-PP1 affects the width of wild type cells. The result in Supp Fig 1e very clearly shows that the drug reduces cell width in wild type cells, almost as much as it reduces the width of *wee1-as* cells. This is a pretty striking phenomenon. I would suggest two small changes in the presentation, because as written this result gets minimal attention. First, the authors should show a direct comparison of cell width between wild type versus *wee1-as* cells, both before and after treatment with inhibitor. This could be obtained by combining graphs in Figure 1c and Supp Fig 1e. Then, they can test if the *wee1-as* mutant has a significant change to cell width when compared to the wild type cells. Second, the authors should be more open in the text about this off-target effect of the drug in wild type cells. The paper is full of excellent data, so it remains an excellent study even though the *wee1-as* cell width phenotype might be partially due to off-target effects on the drug. As written, this control experiment feels brushed aside.

Reviewer #2 (Remarks to the Author):

The authors have performed significant additional experiments to further their work on cell polarization and aspect ratio in *S. japonicus*. Several aspects of my previous comments are well addressed. I remain however concerned about some aspects, in particular that of the unspecific effects of 3BrB-PP1.

Indeed, the effects of 3BrB-PP1 on wt cells, now fully quantified, are rather disturbing. They seem to be almost as strong for cell thinning as in *wee1-as8*, contrary to what the authors state in the text. If you compare the bottom right graph of figure 1b with the right graph of figure S1d, showing the width of the thinner and thicker daughter cell in *wee1-as8* and wt, respectively, the reduction in width is very very similar. I understand that the effect is statistically different in wt and *wee1-as8*, but it remains that 3BrB-PP1 causes important cell thinning independently of inhibition of Wee1. These unspecific effects are similar in *S. pombe* cells, where 3BrB-PP1 causes mild cell thinning independently of the *wee1-as8* mutation. The schemes provided in figure 1b and 2d showing symmetrical wt and asymmetrical *wee1-as8* mutant cells are misleading and should be changed. The data on effects of 3BrB-PP1 in wildtype should be presented in the main figure, where it can be directly compared to that on *wee1-as8*.

The authors also now quantified the dimensions of *wee1-G788E* and *cdc2-G146D* mutants, both of which have only mild cell thinning and aspect ratio significantly lower than wt or *wee1-as8* treated with 3BrB-PP1. Thus, it appears that *S. japonicus* undergoes a mild reduction in cell width upon decrease of cell length, which contributes to maintenance of aspect ratio, but that the complete maintenance of aspect ratio observed upon Wee1-*as8* inhibition with 3BrB-PP1 is a compound response to Wee1 inhibition and off-target effects of the drug that mostly influence cell width. This is actually quite similar to what the authors also describe for *S. pombe*. These findings are interesting and should indeed be reported to challenge the dogma that cell width does not vary. However, the non-specific effect of 3BrB-PP1 should not be underestimated nor under-reported in the text: statements such as (line 128) "resulting in overall conservation of cellular aspect ratio" of *wee1-G788E* cells or (line 130-132) "We observed a broadly similar phenotype in *S. japonicus* cells advanced into

mitosis due to conditional G146D mutation in Cdc2" are inaccurate and exaggerated if you look at figures 1g and S1i, where aspect ratio is significantly lower for both mutants.

The unspecific effects of 3BrB-PP1 also affect the interpretation of other experiments in the manuscript. First, in figure 2, the authors have now quantified the localization of Cdc42 and regulators. However they only provide quantification in cells treated with 3BrB-PP1 and not in untreated cells. Given the off-target effects of 3BrB-PP1 described, this does not give an accurate description of polarity in this organism in the physiological state. Their quantification shows a small hyperpolarization in *wee1-as8* relative to wt (both treated with BrB-PP1), but what is the effect when comparing treated vs. untreated WT or *wee1-as8*? I would not be surprised to see a significant effect too. For instance, the authors state and show that Rga4 normally occupies a small cortical band in the middle of untreated wt cells (see figure S2d), yet upon 3BrB-PP1 treatment, the image shown in figure 2g shows Rga4 distributed over most of the cortex also in wildtype cells. It is a little difficult to compare these two images, as one is a max projection and the other a medial plane image, but it suggests an important effect of 3BrB-PP1 on Rga4 distribution may be unspecific rather than through Wee1 inhibition.

The restoration of medial division in *rga4Δ wee1-G788E* placed in thin channels is very nice and consistent with previous work on division site sliding in round mutants or restoration of microtubule organization in similarly re-shaped cells. It convinces that restoration of elongated aspect ratio helps in division site placement. However, it does not explain the origin of the loss of aspect ratio in the double mutant, which I think is still entirely consistent with the current notion that Rga4 strongly contributes to setting the size of the Cdc42 zone and thus the width of the cell while Wee1 strongly contributes to cell length. What is the width of *rga4Δ* cells untreated with 3BrB-PP1? I do not doubt that Rga4 contributes to setting cell width, but could Rga4 be involved in transducing the cellular response to 3BrB-PP1 independently of Wee1 inhibition?

The asymmetric division is shown only in the case of *wee1-as8* cells treated with 3BrB-PP1, which as highlighted above has strong off-target effects, with wildtype cells also showing asymmetric division. How the more modest width adjustment of *wee1-G788E* and *cdc2-G146D* mutant cells, or that of shift of wildtype cells from rich to poor medium, are made is not investigated. This should be phrased more carefully and also the model figure re-thought.

I think most of my comments can be addressed through text changes and figure modifications. The authors probably already have done the experiments that I ask to be quantified for the localization of polarity factors in untreated cells in figure 2 and for the dimensions of untreated *rga4Δ*.

Minor comments:

In Figure 1b, the data on the y-axis should be the same in both top graphs, yet it isn't. (see for instance the two data point <0.6 on the left graph absent from the right graph)

Why is CRIB quantified after 3h when the other markers are done after 2h in figure 2?

Reviewer #3 (Remarks to the Author):

The authors' revisions and response to reviewers have satisfactorily addressed this reviewer's concerns expressed in the first review of this manuscript, and I recommend it for publication.

In particular, the authors included requested control experiments, explained their reasoning more

clearly, and presented more evidence linking aspect ratio, septum positioning, and viability by confining cells in a microfluidic channel (Fig. 4) and demonstrating binucleation in cells with mis-positioned division sites (Fig. S3e).

Reviewer 1

We are grateful to the reviewer for their suggestions. We edited the manuscript and figures accordingly – please see below.

The authors have addressed my comments and have improved the manuscript nicely. This paper has a number of interesting results and conclusions that are likely to appeal to a broad cell biology audience.

My one lingering concern is the effect of 3-BrB-PP1 on cell width. The authors performed the requested control experiment by testing how 3-BrB-PP1 affects the width of wild type cells. The result in Supp Fig 1e very clearly shows that the drug reduces cell width in wild type cells, almost as much as it reduces the width of *wee1-as* cells. This is a pretty striking phenomenon. I would suggest two small changes in the presentation, because as written this result gets minimal attention. First, the authors should show a direct comparison of cell width between wild type versus *wee1-as* cells, both before and after treatment with inhibitor. This could be obtained by combining graphs in Figure 1c and Supp Fig 1e. Then, they can test if the *wee1-as* mutant has a significant change to cell width when compared to the wild type cells.

We now show combined data in new Fig. 1c and test for significance in cell width changes between 3-BrB-PP1-treated wild type and *wee1-as8* cells. *Wee1* mutants do become thinner and the difference is significant ($p < 0.0001$). Additionally, we combined wild type and *wee1-as8* data to show significant difference in the extent of initial hyperpolarization (new Fig. 1b), as requested by the Reviewer 2.

Second, the authors should be more open in the text about this off-target effect of the drug in wild type cells. The paper is full of excellent data, so it remains an excellent study even though the *wee1-as* cell width phenotype might be partially due to off-target effects on the drug. As written, this control experiment feels brushed aside.

Thank you. We rewrote relevant parts of the text. Hopefully, the revised version provides a clearer description of off-target effects.

Reviewer 2

Many thanks for your comments and suggestions. We have addressed them in full.

The authors have performed significant additional experiments to further their work on cell polarization and aspect ratio in *S. japonicus*. Several aspects of my previous comments are well addressed. I remain however concerned about some aspects, in particular that of the unspecific effects of 3BrB-PP1.

Indeed, the effects of 3BrB-PP1 on wt cells, now fully quantified, are rather disturbing. They seem to be almost as strong for cell thinning as in *wee1-as8*, contrary to what the authors state in the text. If you compare the bottom right graph of figure 1b with the right graph of figure S1d, showing the width of the thinner and thicker daughter cell in *wee1-as8* and wt, respectively, the reduction in width is very very similar. I understand that the effect is statistically different in wt and *wee1-as8*, but it remains that 3BrB-PP1 causes important cell thinning independently of inhibition of *Wee1*. These unspecific effects are similar in *S. pombe* cells, where 3BrB-PP1 causes mild cell thinning independently of the *wee1-as8* mutation. The schemes provided in figure

1b and 2d showing symmetrical wt and asymmetrical *wee1-as8* mutant cells are misleading and should be changed. The data on effects of 3BrB-PP1 in wildtype should be presented in the main figure, where it can be directly compared to that on *wee1-as8*.

The reviewer 1 also requested to combine graphs showing the phenotypes of the wild type and *wee1-as8* cells. As mentioned above, we now show combined data in new Fig. 1b (the extent of initial hyperpolarization) and Fig. 1c (geometric parameters at division). We conclude that Wee1 mutants become thinner and the difference is significant ($p < 0.0001$).

The authors also now quantified the dimensions of *wee1-G788E* and *cdc2-G146D* mutants, both of which have only mild cell thinning and aspect ratio significantly lower than wt or *wee1-as8* treated with 3BrB-PP1. Thus, it appears that *S. japonicus* undergoes a mild reduction in cell width upon decrease of cell length, which contributes to maintenance of aspect ratio, but that the complete maintenance of aspect ratio observed upon Wee1-*as8* inhibition with 3BrB-PP1 is a compound response to Wee1 inhibition and off-target effects of the drug that mostly influence cell width. This is actually quite similar to what the authors also describe for *S. pombe*. These findings are interesting and should indeed be reported to challenge the dogma that cell width does not vary. However, the non-specific effect of 3BrB-PP1 should not be underestimated nor under-reported in the text; statements such as (line 128) “resulting in overall conservation of cellular aspect ratio” of *wee1-G788E* cells or (line 130-132) “We observed a broadly similar phenotype in *S. japonicus* cells advanced into mitosis due to conditional G146D mutation in *Cdc2*” are inaccurate and exaggerated if you look at figures 1g and S1i, where aspect ratio is significantly lower for both mutants.

The reviewer is correct – the difference in width changes between 3-BrB-PP1-treated *wee1-as8* and *wee1-ts* and *cdc2-ts* cells shifted to the restrictive temperature is approximately 50%. In *wee1-ts* cells the aspect ratio does not deviate far from the controls (Fig. 1g). It does drop further in *cdc2-ts* cells (Supplementary Fig. 1h-i). We edited the manuscript accordingly. Of note, shifting *S. japonicus* cells from YES to EMM medium does cause a much more pronounced change in cell diameter (~1 micron).

The unspecific effects of 3BrB-PP1 also affect the interpretation of other experiments in the manuscript. First, in figure 2, the authors have now quantified the localization of Cdc42 and regulators. However they only provide quantification in cells treated with 3BrB-PP1 and not in untreated cells. Given the off-target effects of 3BrB-PP1 described, this does not give an accurate description of polarity in this organism in the physiological state. Their quantification shows a small hyperpolarization in *wee1-as8* relative to wt (both treated with BrB-PP1), but what is the effect when comparing treated vs. untreated WT or *wee1-as8*? I would not be surprised to see a significant effect too. For instance, the authors state and show that Rga4 normally occupies a small cortical band in the middle of untreated wt cells (see figure S2d), yet upon 3BrB-PP1 treatment, the image shown in figure 2g shows Rga4 distributed over most of the cortex also in wildtype cells. It is a little difficult to compare these two images, as one is a max projection and the other a medial plane image, but it suggests an important effect of 3BrB-PP1 on Rga4 distribution may be unspecific rather than through Wee1 inhibition.

As requested, we now show and quantitate three experimental conditions in new Fig. 2: (1) wild type cells in solvent control (to illustrate Cdc42 polarity in the ‘physiological state’ in *S. japonicus*); (2) wild type cells in the presence of 3-BrB-PP1 (a control for Wee1 inhibition, as suggested during previous

revision); and (3) *wee1-as8* cells in the presence of 3-BrB-PP1. We decided to include estimations of the extent of tip polarization for CRIB-3xGFP, Scd2-mNeonGreen, Gef1-mNeon Green and Scd1-mNeonGreen using two approaches. First, we show tip-to-tip ratio (new Fig. 2d). Second, we show 'polarization index' (average intensity of tip fluorescence normalized to average intensity of entire cortex, shown in Supplementary Fig. 2d). Both metrics report on polarization but each has its strengths and limitations, and we think that, together, they provide a more accurate way of viewing polarity. The tip-to tip ratio is reporting the extent of asymmetry in polarization. The 'polarization index' reports on relative tip enrichment of cellular fluorescence but it is sensitive to cell length. Both methods confirm that 3-BrB-PP1 treatment of *wee1-as8* cells results in increase in polarization for CRIB, Scd2 and Gef1, as compared to the control.

Interestingly, treatment of wild type cells with 3-BrB-PP1 did not produce significant enrichment of CRIB or Scd2 at the tips, even though cells did thin out. We did observe some 3-BrB-PP1-related enrichment of Gef1, however, it was concentrated further in 3-BrB-PP1-treated *wee1-as8* cells. Of note, we repeated the Gef1 experiment several times since Gef1 tip enrichment has been linked to cellular stress in *S. pombe* (by the Sawin lab), and we are confident that we are reporting the actual state of polarity for this protein.

It is worth saying that Rga4 clearance from the growing tip was specific to Wee1 inhibition – we did not observe any differences between the solvent control and 3-BrB-PP1-treated wild type cells. As can be seen from the single medial plane image, distribution of Rga4-mNeonGreen is similar between the two conditions. We removed the maximum projection images of Rga4 in *S. japonicus* vs *S. pombe* from the Supplementary Fig. 2, as those can be perhaps somewhat misleading. Indeed, Rga4 appears much more concentrated at the cell middle in maximum projection images in *S. japonicus*, but this could be because of very high intensity of medial nodes in this organism.

The restoration of medial division in *rga4Δ wee1-G788E* placed in thin channels is very nice and consistent with previous work on division site sliding in round mutants or restoration of microtubule organization in similarly re-shaped cells. It convinces that restoration of elongated aspect ratio helps in division site placement. However, it does not explain the origin of the loss of aspect ratio in the double mutant, which I think is still entirely consistent with the current notion that Rga4 strongly contributes to setting the size of the Cdc42 zone and thus the width of the cell while Wee1 strongly contributes to cell length. What is the width of *rga4Δ* cells untreated with 3BrB-PP1? I do not doubt that Rga4 contributes to setting cell width, but could Rga4 be involved in transducing the cellular response to 3BrB-PP1 independently of Wee1 inhibition? I think most of my comments can be addressed through text changes and figure modifications. The authors probably already have done the experiments that I ask to be quantified for the localization of polarity factors in untreated cells in figure 2 and for the dimensions of untreated *rga4Δ*.

We analyzed the data and the reviewer is correct – *rga4Δ* cells do not become significantly thinner in the presence of 3BrB-PP1 (new Supplementary Fig. 3b). However, they are also deficient in scaling in other conditions tested: *wee1-G788E* at 30°C (Fig. 4) and upon shift to the minimal medium

(Supplementary Fig. 3). Taken together, we conclude that whereas *S. japonicus* cells can maintain linear growth in the absence of Rga4 (with cells being slightly wider, similar to the situation in *S. pombe*), this GAP is required for dynamic modulation of cell width in a number of scenarios. Here we use Rga4 deficiency as a handle on cellular geometry scaling.

The asymmetric division is shown only in the case of *wee1-as8* cells treated with 3BrB-PP1, which as highlighted above has strong off-target effects, with wildtype cells also showing asymmetric division. How the more modest width adjustment of *wee1-G788E* and *cdc2-G146D* mutant cells, or that of shift of wildtype cells from rich to poor medium, are made is not investigated. This should be phrased more carefully and also the model figure re-thought.

Both *wee1-G788E* and *cdc2-G146D* cells actually go through an asymmetric stage during the temperature shift-up. We did not include this data into our original submission because of space issues. We now show 3-hour time points for both *wee1* (Supplementary Fig. 1g) and *cdc2* (Supplementary Fig. 1h, middle panel) mutants. There are clearly asymmetrically dividing cells in both cultures – in fact, one can still see them in a 6-hour time point for *cdc2-G146D*. We used *wee1-G788E* later in the paper to show that 1) Rga4 is required for scaling; and 2) that physically constraining cell width rescues division site positioning in cells unable to scale (Fig. 4).

Similarly, wild type cells shifted to EMM go through asymmetric divisions, as can be seen in Fig. 1h (4- and 7-hour time points).

We edited the model figure so that it is more precise – we now only mention ‘Cell width adjustment, dependent on Cdc42 GAP Rga4’.

Minor comments:

In Figure 1b, the data on the y-axis should be the same in both top graphs, yet it isn't. (see for instance the two data point <0.6 on the left graph absent from the right graph)

Thank you very much for spotting this! A mix-up on our part, now corrected.

Why is CRIB quantified after 3h when the other markers are done after 2h in figure 2?

We repeated this experiment for the 2-hour time point, as shown for all other markers. It was really a historical reason – that was the first marker we analyzed.

Reviewer 3

The authors' revisions and response to reviewers have satisfactorily addressed this reviewer's concerns expressed in the first review of this manuscript, and I recommend it for publication.

In particular, the authors included requested control experiments, explained their reasoning more clearly, and presented more evidence linking aspect ratio, septum

positioning, and viability by confining cells in a microfluidic channel (Fig. 4) and demonstrating binucleation in cells with mis-positioned division sites (Fig. S3e).

Thank you very much for your encouraging comments.